# Endogenous Hormone Levels and Transcriptomic Analysis Reveal the Mechanisms of Bulbil Initiation in *Pinellia ternata*

**DOI:** 10.3390/ijms25116149

**Published:** 2024-06-03

**Authors:** Lan Mou, Lang Zhang, Yujie Qiu, Mingchen Liu, Lijuan Wu, Xu Mo, Ji Chen, Fan Liu, Rui Li, Chen Liu, Mengliang Tian

**Affiliations:** College of Agronomy, Sichuan Agricultural University, Chengdu 611130, China; 0350magnolia@gmail.com (L.M.); sj820937834@gmail.com (L.Z.); 2020101011@stu.sica.edu.cn (Y.Q.); 15033515860@163.com (M.L.); lichuanwu1990@gmail.com (L.W.); 2020301094@stu.sicau.edu (X.M.); jichen@sicau.edu.cn (J.C.); 10060@sicau.edu.cn (F.L.); 71352@sicau.edu.cn (R.L.); 71147@sicau.edu.cn (C.L.)

**Keywords:** *Pinellia ternata*, bulbil, hormone, transcriptomics, metabolomics

## Abstract

*Pinellia ternata* is a medicinal plant that has important pharmacological value, and the bulbils serve as the primary reproductive organ; however, the mechanisms underlying bulbil initiation remain unclear. Here, we characterized bulbil development via histological, transcriptomic, and targeted metabolomic analyses to unearth the intricate relationship between hormones, genes, and bulbil development. The results show that the bulbils initiate growth from the leaf axillary meristem (AM). In this stage, jasmonic acid (JA), abscisic acid (ABA), isopentenyl adenosine (IPA), and salicylic acid (SA) were highly enriched, while indole-3-acetic acid (IAA), zeatin, methyl jasmonate (MeJA), and 5-dexoxystrigol (5-DS) were notably decreased. Through OPLS-DA analysis, SA has emerged as the most crucial factor in initiating and positively regulating bulbil formation. Furthermore, a strong association between IPA and SA was observed during bulbil initiation. The transcriptional changes in *IPT* (*Isopentenyltransferase*), *CRE1* (*Cytokinin Response 1*), *A-ARR* (*Type-A Arabidopsis Response Regulator*), *B-ARR* (*Type-B Arabidopsis Response Regulator*), *AUX1* (*Auxin Resistant 1*), *ARF* (*Auxin Response Factor*), *AUX/IAA* (*Auxin/Indole-3-acetic acid*), *GH3* (*Gretchen Hagen 3*), *SAUR* (S*mall Auxin Up RNA*), *GA2ox* (*Gibberellin 2-oxidase*), *GA20o*x (*Gibberellin 20-oxidase*), *AOS* (*Allene oxide synthase*), *AOC* (*Allene oxide cyclase*), *OPR* (O*xophytodienoate Reductase*), *JMT* (*JA carboxy l Methyltransferase*), *COI1* (*Coronatine Insensitive 1*), *JAZ* (*Jasmonate ZIM-domain*), *MYC2* (*Myelocytomatosis 2*), *D27* (*DWARF27*), *SMAX* (*Suppressor of MAX2*), *PAL* (*Phenylalanine Ammonia-Lyase*), *ICS* (*Isochorismate Synthase*), *NPR1* (*Non-expressor of Pathogenesis-related Genes1*), *TGA* (*TGACG Sequence-specific Binding*), *PR-1* (*Pathogenesis-related*), *MCSU* (M*olybdenium Cofactor Sulfurase*), *PP2C* (*Protein Phosphatase 2C*), and S*nRK* (*Sucrose Non-fermenting-related Protein Kinase 2*) were highly correlated with hormone concentrations, indicating that bulbil initiation is coordinately controlled by multiple phytohormones. Notably, eight TFs (transcription factors) that regulate AM initiation have been identified as pivotal regulators of bulbil formation. Among these, *WUS* (*WUSCHEL*), *CLV* (*CLAVATA*), *ATH1* (*Arabidopsis Thaliana Homeobox Gene 1*), and *RAX* (R*egulator of Axillary meristems*) have been observed to exhibit elevated expression levels. Conversely, *LEAFY* demonstrated contrasting expression patterns. The intricate expression profiles of these TFs are closely associated with the upregulated expression of *KNOX*(*KNOTTED-like homeobox*), suggesting a intricate regulatory network underlying the complex process of bulbil initiation. This study offers a profound understanding of the bulbil initiation process and could potentially aid in refining molecular breeding techniques specific to *P. ternata*.

## 1. Introduction

*Pinellia ternata* is a perennial herb of the Araceae family [1]. Its tuber contains a diverse array of bioactive metabolites, such as alkaloids, cardiac glycosides, sterols, aromatic acids, fatty acids and esters, polysaccharides, volatile oils, amino acids, pinellia proteins, and inorganic elements, which exhibit various pharmacological effects, e.g., expectorant activity and potential therapeutic benefits against tumors, inflammation-related disorders, ulcers, and early pregnancy [2]. *P. ternata* has been extensively utilized in China for centuries, and its medicinal properties were first described in *Shen Nong’s Herbal Classic* [3]. However, in recent years, wild resources of *P. ternata* have been severely depleted due to unregulated mining practices and herbicide misuse [4]. Moreover, *P. ternata* faces the obstacle of continuous cropping during artificial planting [5]. Therefore, suitable artificial cultivation methods must be established to ensure the sustainable conservation and utilization of *Pinellia* resources.

*P. ternata* can be propagated through seeds, bulbils, and tubers [4]. At least three years are required to reach maturity for harvesting if seed propagation is used, while harvesting is possible in the second year when bulbil propagation is adopted [6]. Previously, propagation with bulbils has shown a high survival rate and fast germination cycles [5]. In addition, the yield increase achieved through tuber propagation is not as substantial as that achieved through bulbil propagation [6]. Since bulbils represent the most favorable propagules for *P. ternata*, investigating their occurrence and developmental mechanisms holds paramount importance for ensuring sustainable resource utilization. In terms of cultivation time and yield, bulbils are the best breeding material for *P. ternata.*

Bulbils fulfil a dual role as both agents of reproduction and sources of nourishment; they typically emerge in specific parts of plants, such as leaf axils, stems, or inflorescences, primarily serving as a means of asexual reproduction [7]. Numerous medicinal species, e.g., *Dioscorea opposita* and *Lilium sulphureum*, produce bulbils on their leaves and petioles [8]; conversely, plants such as *Agave tequilana* and *Polygonum viviparum* develop bulbils on their inflorescences [9,10], and bulbils can also grow on the leaves of plants such as *Pinellia cordata*, *Amorphophallus bulbifer*, and *Ornithogalum thyrsoide* [11]. In *P. ternata*, each tuber can produce one to several compound leaves, and each compound leaf typically grows with a bulbil at the base of its petiole [11]. However, an additional bulbil occasionally develops at the top of the petiole, resulting in two bulbils growing on a single one [12]. Therefore, in *P. ternata*, bulbils grow on the petioles with diverse phenotypes. The bulbils are generally spherical or oval, though other irregular shapes also occur, depending on the specific characteristics and physiological state of the plant [7]. When bulbils grow at both the top and base of the petiole, they are termed double-bulbil type (DB); when bulbils only grow at the base of the petiole and the top of the petiole does not grow bulbils, they are termed single-bulbil type (SB). The bulbil phenotype of *P. ternata* is genetically stable [12]. With the aims of alleviating the resource crisis facing *P. ternata* and enhancing its reproductive efficiency, exploring the mechanisms of bulbil initiation has become paramount in breeding efforts. Understanding these mechanisms can provide a theoretical framework for developing effective breeding methods. In this study, we utilized SB as a control to investigate the process of top bulbil initiation in DB *P. ternata*. This investigation aims to shed light on the general mechanisms of bulbil initiation in *P. ternata*, paving the way for future breeding strategies.

The bulbils of *P. ternata* are hypothesized to initiate from the axillary bud of the leaf AM, and the formation of these abnormal axillary buds is regulated by the shoot apical meristem and the lateral meristem [13,14]. The AMs in plants originate from the apical meristem near the stem tip. Simultaneously, a boundary forms between the main stem and the leaf primordia, which separates the group of meristematic cells from the developing organs. Within this boundary, cell division is decreased, the expression of cell cycle-related genes is downregulated, and the cell wall becomes relatively rigid [15,16]. The formation of this boundary is regulated by network control genes, such as *CUC* (*Cup-shaped Cotyledon*), *STM* (*Shoot Meristemless*), and *LOB* (*Lateral Organ Boundaries*), which suppress cell division and differentiation and regulate the spatial distribution of endogenous hormones such as auxin, GAs, and CTKs. Many studies have shown that the genes expressed within the boundary are essential for the formation of AMs [17,18]. Loss or downregulation of the expression of these genes leads to organ fusion and AM defects, whereas their overexpression results in the ectopic formation of AMs [19,20]. TFs are highly conserved across species, some of which play crucial roles in multiple developmental processes, demonstrating the importance of regulatory networks in meristem development [21].

To date, the molecular regulatory mechanisms underlying the development of plant bulbils have not been thoroughly elucidated. Several genes controlling diverse aspects of bulbil development have been documented in different species. For example, WUS-related homeobox (WOX) proteins constitute a plant-specific family of eukaryotic homeobox TFs, which play crucial roles in promoting cell division, inhibiting cellular differentiation, regulating embryonic development, maintaining stem cell niches in the meristem, and facilitating organ formation [22,23]. Notably, two specific *WOX* genes, *LlWOX9* and *LlWOX11*, were associated with bulbil formation in *Lilium lancifolium* [24]. *AGO1* (*Argonaute RISC component 1*) is a pivotal gene involved in the regulatory pathway of microRNAs (miRNAs), and it plays a crucial role in orchestrating cellular division and differentiation during tissue development [25]. The *LIAGO1* gene of *Lilium lancifolium* has exhibited an association with bulbil formation [8]. MADS-box family genes encode TFs that are involved in multiple developmental processes in plants, especially floral organ specification, fruit development, and ripening [26]. In *Agave tequilana*, the *MADS* (*Multiple Adenosine Dinucleotide Sequences*) genes were associated with bulbil development [27]. *KNOX*, a TF with homologous and heterogeneous domains, is essential for the formation and functional maintenance of meristem tissues during organ morphogenesis [28]. During the bulbil formation of *A. tequilana*, *AtqKNOX1* was significantly upregulated in the initial stage, while the expression of *AtqKNOX2* increased during later stages [29]. In *Dioscorea alata*, several TFs from Aux/IAA, ERF (Ethylene-Responsive Factor), MYB (MYB-related) and bHLH (basic helix-loop-helix) families have been shown to play key roles in triggering bulbil formation [7].

Despite the above knowledge, little is known about the genetic regulation of bulbil formation in *P. ternata*. In angiosperms, the PEBP (Phosphatidylethanolamine-Binding) gene family plays an important role in organ development, e.g., floral initiation and underground storage organ formation [30]. The transcriptome and profiles of *Pinellia cordata* were analyzed, and a total of fourteen unigenes homologous to PEBP gene family members were identified; there were four unigenes with differential expressions in three kinds of tissue (petiole, bulbil, and tuber), as revealed by the profiles, which indicates that members of the PEBP gene family could be involved in the bulbil development of *Pinellia cordata* [31]. However, the roles of many other genes are elusive. To explore the role of different hormones and gene expressions in the development of bulbils, we hypothesize that bulbil development could be related to a complex regulatory network; therefore, this study aims to employ targeted metabolomics combined with transcriptomic analysis to investigate changes in gene expression and hormone levels during the initiation of *P. ternata* bulbils. We aim to unearth the molecular regulatory network and explore the molecular mechanisms underlying *P. ternata* bulbil development.

## 2. Results

### 2.1. Phenotypic and Histomorphological Analyses

Naked eye observation reveals that the development of the top bulbil in DB *P. ternata* can be divided into three distinct stages: initiation, expansion, and maturity. Initially, the bulbil appears as a white dot bulge, which gradually expands and matures, ultimately developing into a brown, elliptical sphere (Figure 1a–c). Figure 1d–f show CT scanning images of the apical bulbil transverse section at its three different developmental stages, clearly showing the sequential alteration of its internal structure. Figure 1g–i show CT scanning images of the apical bulbil longitudinal section at its three different developmental stages, clearly showing the sequential alteration of its internal structure. By integrating 2D scanning images captured from diverse angles, we have successfully simulated 3D images of the bulbil during its three distinct developmental stages (Figure 1j–l). The apical bulbil differentiated from the parenchyma cells located in the leaf axils at the top of the petiole. The 2D images showed that the development of the bulbil was a process of initial longitudinal growth before lateral expansion. In the mature stage, several layers of scales on the top of the bulbil wrap around the apical meristem, with a hollow structure inside, and large amounts of nutrients accumulate at the bottom of the bulbil.

To establish a precise timeline for the initial germination of the top bulbil, we conducted a detailed comparative analysis of the morphological developmental disparities between DBs and SBs using paraffin sections. We found that for the DB and SB in the early stages of development, there was no obvious difference in the top bulbil. Buds emerged after the tubers were cultured in an illuminated incubator for 5–7 days, and several adventitious roots sprouted (Figure 2a,m). After another 3–5 days, opaque tissue, which was the early primordium of the base bulbil, appeared at the base petiole (Figure 2d,p). As the base primordium evolved into a white early bulbil and became enlarged, the petiole and leaf also grew rapidly. The ventral surface of the young leaves presented a downward and inward curling state (Figure 2g,s). However, with the elongation of the petiole and development of the leaves, the blades slowly moved upward and spread, and the ventral surface on the top of the petiole was gradually revealed. At this point, a small white spot was observed on the top of the DB petiole, whereas it was not found in the SB petiole. At this stage, the external morphological features of the DB and SB exhibited significant differences (Figure 2j,v).

During the tuber sprouting stage and basal bulbil initiation stage, no signs of top bulbil primordium initiation were observed, and no differences in the tissue sections were detected between the SB and DB petioles (Figure 2b,c,e,f,n,o,q,r). However, in the DB petiole, prior to leaf spreading, some parenchymal cells increased in size and protruded from the top of the petiole, and some vascular bundles protruded towards the bulbil-protruding direction in the DB petiole (Figure 2t,u). In the SB, prior to leaf spreading, no parenchymal cells increased in size or protruded (Figure 2h,i). The red arrow designates the location where the top bulbil germinated in the DB, whereas the green arrow designates the location where no parenchymal hyperplasia occurred in SB during the same period (Figure 2i,u). Based on the above data, we conclude that the initiation of the top bulbil formation occurred before the leaf completely unfolded. Therefore, we chose this stage to collect transcriptome and metabolome samples.

### 2.2. Phytohormone Analysis

A comparative metabolome analysis was performed between the DUs and SUs to investigate hormonal changes during top bulbil initiation, the quantitative hormone findings are shown in Appendix A, and the raw data are shown in Appendix A. In the early stage of top bulbil initiation, the content of IAA, IBA, IP, zeatin, MeJA, and 5-DS tended to decrease, while IPA, SA, ABA, TZR, and JA showed increasing trends. The content of MESA remained unaltered. In PCA, the metabolites were clearly separated between the two cultivars (Figure 3A). PC1 and PC2 accounted for 91.4% and 5.2% of the variance, respectively. The OPLS-DA results demonstrate significant differences between SU and DU, with all six samples falling within the 95% confidence interval, indicating clear clustering of the biological replicates within the same group (Figure 3B). The UV-formatted data that were generated using SIMCA software (V16.0.2) yielded R2Y and Q2 values of 0.999 and 0.99 cum, respectively, indicating the effectiveness of the OPLS-DA model (Figure 3C). Eight differentially accumulated hormones were selected based on a VIP ≥ 1 and *p* ≤ 0.05; JA, ABA, IPA, and SA in the DUs were higher than in the SUs, while IAA, zeatin, MeJA, and 5-DS were lower (Appendix A). The volcano plot (Figure 3D) shows that SA was the most critical for the initiation of bulbil formation, positively regulating the initiation. IPA, ABA, JA, and IAA were also important for bulbil regulation.

Based on the raw data, we computed an Euclidean distance matrix to quantify metabolite values, and we performed hierarchical clustering analysis (HCA) using the complete linkage method to generate a heatmap illustrating the results of the SU vs. DU (Figure 4A). A matchstick analysis revealed significant changes in the levels of SA, IPA, ABA, and JA (Figure 4B). By calculating Pearson correlation coefficients for the metabolite levels, we identified a strong association between IPA and SA during top bulbil initiation (Figure 4C).

### 2.3. Transcriptome Assembly Analysis

To elucidate the genetic factors underlying top bulbil initiation, we conducted a comparative transcriptomic analysis of SU and SM from SB, and DU, and DM from DB, with three biological replicates for each sample (Figure 5A). After enforcing stringent quality control measures, we obtained an average of 22,405,460 (6.73 Gb) clean Illumina reads for SU and SM, along with 21,626,296 (6.5 Gb) clean reads for DU and DM. The Q20 and Q30 scores surpassed 96% and 91%, respectively, for both the SB and DB samples, and the GC content exceeded 52% (Appendix A). Subsequently, Trinity was used to de novo assemble a transcriptome based on the filtered clean reads, which generated 125,645 unigenes with an average length of 805 bp (shortest = 301 bp; longest = 16,487 bp). The N50 length of this assembly reached 993 bp (Figure 5B), while the majority of unigenes (54,554, 43.4%) had sequence lengths within the range of 300–500 bp, followed by 500–1000 bp (42,959, 34.2%) and 1–2 kb (20,671, 16.5%).

### 2.4. Functional Annotation and Classification

MDS plot among each sample was obtained by using the MDS analysis method (Figure 5C). It is evident from the plot that the degree of similarity among individual sample groups is notably high. Furthermore, a distance is observable between DUs and SUs, indicating a significant divergence in their gene expression profiles. This outcome indicates that further analysis of the difference in gene expression is warranted. The assembled 125,645 transcripts were individually aligned to the KEGG and GO databases using BLASTX. In the GO annotations, 36,854 (29.33%) unigenes were assigned to biological processes, cellular components, or molecular function categories (Appendix A). The KO annotations and KEGG pathway classifications revealed that in DU vs. SU the majority of the unigenes were involved in plant–pathogen interactions, followed by starch and sucrose metabolism and plant hormone signal transduction (Figure 5D).

### 2.5. Analysis of Differentially Expressed Genes

The results of the pairwise comparisons between the SB and DB libraries are presented in Appendix A. The transcriptomic comparisons of SB vs. DB identified 9112, 3630, 1021, and 7552 DEGs in the DU vs. SU, DU vs. DM, SU vs. SM, and DM vs. SM groups, respectively (Figure 6A). Among them, 4573 had higher expression levels in DUs than in SUs, 3201 had higher expression in DUs than in DMs, 695 had higher expression in SUs than in SMs, and 3409 had higher expression in DMs than in SMs (Figure 6A). Figure 6B shows the common and unique differential genes of the four samples. There were a total of 35 DEGs in the four comparison groups, and there were varying amounts of DEGs in every two or three comparison groups. There were 3333, 1215, 3465, and 241 DEGs in DM vs. SM, DU vs. DM, DU vs. SU, and SU vs. SM comparison groups, respectively. We focused on the differences between DU and SU during the same growth period, and the DEGs were listed in Appendix A. To account for sample background differences between the DU vs. SU and between DM vs. SM groups, the DEGs identified from the comparison of the DU and SU were subtracted from those identified from the DM vs. SM, resulting in a total of 5403 DEGs, including 3154 upregulated and 2249 downregulated genes (Figure 6C). In the KEGG enrichment analysis, DEGs involved in hormone signal transduction pathways were significantly enriched (Figure 6D).

### 2.6. Analysis of Differentially Expressed Genes Involved in Hormone Synthesis and Signaling Pathways

The metabolome analysis indicated significant differences in the levels of JA, ABA, IPA, and SA between the DUs and SUs, and higher levels of IAA, zeatin, MeJA, and 5-DS were observed in the SUs than in the DUs. We further analyzed the DEGs involved in hormone biosynthesis and signaling pathways, including the 5-DS, CTK, JA, ABA, IAA, GA, and SA pathways (Figure 7 and Figure 8, Appendix A).

Through transcriptome analysis, we identified one *D27*, four *D14* (*DWARF14*), one *SMAX*, and two *CYP711A1* unigenes in the 5-DS synthesis and signaling pathway. The expression of *D27* (Cluster-8921.57758) was twofold lower in the DUs than in the SUs. And *SMAX* (Cluster-8921.40244) was higher in the DUs compared to SUs. The expression levels of two *D14* unigenes (Cluster-8921.72552, Cluster-8921.61766) in the DUs were more than double those in the SUs. The *D14* (Cluster-8921.52346) unigene in the DUs exhibited an expression level that was one-third of that in the SUs, while another *D14* (Cluster-8921.27440) unigene was undetectable in the SUs. The expression of two *CYP711A1* unigenes, one (Cluster-8921.79128) was upregulated and another (Cluster-8921.6134) was downregulated in the DUs compared to SUs.

In the CTK biosynthesis and signaling pathways, we identified that two *IPT*, two *CRE1*, two *AHP*, one *A-ARR* and two *B-AR*R were differentially expressed in DUs vs. SUs. Among them, all of the *CRE1* (cluster-8921.49077, cluster-8921.24392), *A-ARR* (cluster-8921.41617, cluster-8921.67255), and *B-AR*R (cluster-8921.85010) were upregulated, while *AHP* (cluster-8921.73356, cluster-8921.19182) were downregulated. The expression levels of *IPT* (Cluster-8921.37002) was elevated and another *IPT* (Cluster-8921.7088) was downregulated in DUs compared to SUs.

In the JA biosynthesis and signaling pathways, we identified that three *LOX*, five *AOS*, six *AOC*, one *JMT*, one *OPR*, one *COI1*, four *JAZ*, and one *MYC2* unigenes were differentially expressed in DUs vs. SUs. In JA biosynthesis pathway, the expression levels of *AOS* (Cluster-8921.58118, Cluster-8921.60469, Cluster-8921.86856, Cluster-8921.36662, Cluster-8921.25905), *AOC* (Cluster-8921.66780, Cluster-8921.55723, Cluster-8921.80501, Cluster-8921.52264, Cluster-8921.70680, Cluster-8921.52262), *OPR* (Cluster-8921.23729), and *JMT* (Cluster-8921.57682) were greater in the DUs than in the SUs, and there were two *LOX* (Cluster-8921.7181, Cluster-8921.83767) downregulated, while one *LOX* (Cluster-8921.63135) upregulated in the DUs compared to the SUs. In the JA signaling pathway, the expression level of *COI1* (cluster-8921.62063) was downregulated in the DUs compared to the SUs, whereas the expression levels of *JAZ* (cluster-8921.63700, cluster-8921.64422, cluster-8921.51347, cluster-8921.69647) and *MYC2* (cluster-8921.53089) were greater in the DUs than in the SUs.

During the initiation of *P. ternata* bulbils, there were four *NCED* (*Neural Ectodermal Cell Differentiation*), one *MCSU*, two *PP2C*, one *SnRK2*, and two *ABF* (*Abscisic Acid-Responsive Element Binding Facto*r) DEGs were implicated in the ABA biosynthesis and signaling pathways. Three *NCED* (Cluster-8921.54218, Cluster-8921.75963, Cluster-8921.87166) were upregulated, while one *NCED* (Cluster-8921.88400) and *MCSU* (Cluster-8921.55329) were downregulated in the ABA biosynthesis pathway. In the ABA signaling pathway, the expression levels of *PP2C* (cluster-8921.50405, cluster-8921.57559) and *ABF* (cluster-8921.53684) were elevated, whereas the expression of *SnRK2* (cluster-8921.54876) and *ABF* (cluster-8921.4832) were downregulated in the DUs compared to the SUs.

In this study, we identified one *YUCCA* (*YUCCA Flavin Monooxygenas*), two *TDC* (*tryptophan decarboxylase*), four *ALDH* (*Aldehyde Dehydrogenase*), one *AUX1*, four A*UX/IAA*, three *ARF*, thirteen *SAUR*, and one *GH3* as differentially expressed genes in the Auxin biosynthesis and signaling pathways. Among them, *YUCCA* (Cluster-8921.104070) and *AUX/IAA* (cluster-8921.63362, cluster-8921.60258, cluster-8921.18389, cluster-8921.57684), ARF (cluster-8921.86728), and *SAUR* (cluster-8921.59122, cluster-8921.3740, cluster-8921.104731, cluster-8921.94159, cluster-8921.37081, cluster-8921.88733, cluster-8921.80561) were downregulated, whereas *TDC* (Cluster-8921.50137, Cluster-8921.80448), *ALDH* (Cluster-8921.52329, Cluster-8921.76406, Cluster-8921.78519, Cluster-8921.31990), *AUX1* (cluster-8921.41248), *ARF* (cluster-8921.17746, cluster-8921.38353), *SAUR* (cluster-13514.0, cluster-8921.47148, cluster-8921.50983, cluster-8921.73014, cluster-8921.52060, cluster-8921.89074), and *GH3* (cluster-8921.94465) were upregulated in bulbil initiation.

In the GA biosynthesis and signaling pathways, we identified that two *GA20ox*, six *GA2ox*, *three DELLA* (*DELLA domain*), and six *TF* were differentially expressed in DUs vs. SUs. The transcriptome sequencing revealed that the expression levels of three *GA2ox* (Cluster-8921.54748, Cluster-8921.53834, Cluster-8921.39441, Cluster-8921.69332, Cluster-8921.87643, Cluster-8921.87642), one of *GA20ox* (Cluster-8921.93746), *DELLA* (cluster-8921.86409, cluster-8921.82815, cluster-8921.82816), and two *TF* (cluster-8921.81801, cluster-8921.60334) unigenes were elevated in bulbil initiation, and four *TF* (cluster-8921.51044, cluster-8921.75269, cluster-8921.67736, cluster-8921.62359) and one of GA20ox (Cluster-8921.27053) were downregulated. 

In the SA biosynthesis and signaling pathways, we identified one *ICS*, three *4CL* (*4-Coumarate Coenzyme A Ligase*), one *PAL*, three *NPR1*, two *TGA*, and three *PR-1* were differentially expressed in DUs vs. SUs. During the initial stage of bulbil initiation in *P. ternata*, the expressions of the *ICS* (Cluster-8921.78302), *PR-1* (cluster-8921.32229, cluster-8921.57094, cluster-8921.57095), and one of *4CL* (Cluster-8921.56507) were downregulated, whereas the expressions of the *PAL* (Cluster-8921.75594), *NPR-1* (cluster-8921.67167, cluster-8921.55488, cluster-8921.48039), *TGA* (cluster-8921.48480, cluster-8921.81164), and two *4CL* (cluster-8921.48480, cluster-8921.81164) unigenes exhibited an upregulated trend.

### 2.7. DEGs Involved in the AM Initiation Regulatory Network

According to the genes involved in AM development in *Oryza sativa*, *Arabidopsis thaliana*, *Lycopersicon esculentum*, *Zea mays*, and other plants [32], we identified 12 key genes associated with the regulation of AM development from the transcriptome data of *P. ternata* (Appendix A). The expression levels of *KNOX*, *WUS*, *CLV*, *ATH1*, *SRL*, *MOC1*, *CUC*, *RAX*, *REV* (*Revolute*), and *SPL* (*SQUAMOSA-promoter binding protein-like*) were significantly greater in the DUs than in the SUs, indicating their upregulation during top bulbil differentiation. Additionally, *LEAFY* and *DA1* showed higher expression levels in the SUs, suggesting their decreased expression during this developmental stage.

### 2.8. Verification of Gene Expression

We conducted qRT–PCR validation to assess the expression levels of selected DEGs involved in hormone biosynthesis and signaling pathways, as well as in the regulation of AM formation (Appendix A). The relative expression levels of the selected candidate genes can be divided into three categories. The first category contains *KNOX*, *WUS*, *CLV*, *CUC*, *GA20ox*, *A-ARR*, *B-ARR*, *RAX*, *CRE1*, *IPT*, *ARF*, and *AUX1*, which were all highly expressed in the DUs; of particular note is *KNOX*, which increased during the period of bulbil initiation and was hardly expressed in other sites. The second category comprises *GH3*, which was highly expressed in the DMs and lowly expressed in the SUs. The third category includes *GA2ox* and *AUX/IAA*, which were highly expressed in the SUs. At the same time, a comparison between the expression patterns resulting from qPCR and RNA-seq was performed (Appendix A). The RNA-seq expression patterns of the 12 genes classified into the first category in qPCR analysis are consistent with the qPCR results, and all of the genes showed high expression in the DUs. The RNA-seq expression pattern of *GH3*, classified into the second category, was DM > DU > SM > SU, which is also consistent with the qPCR results. The RNA-seq expression pattern of *AUX/IAA*, classified into the third category, was SU > DU > DM > SM, and the RNA-seq expression pattern of *GA2ox* was SU > SM > DU > DM, which is also consistent with the qPCR results. This further proves the accuracy of the RNA-seq data.

## 3. Discussion

### 3.1. Morphological and Histological

The development stages of plant bulbils generally include initiation, growth, differentiation, and maturation [33]. In this study, we observed that the development of the top bulbil of the DB also went through these stages, and the top bulbil’s initiation occurred prior to complete leaf unfurling. This conclusion provides a sampling time frame for studying the molecular mechanisms of *P. ternata* bulbil initiation. Different plants exhibit varying developmental pathways for their bulbils. For example, bulbils morphologically progress from a ‘torpedo shape’ to an ‘embryo shape’ and finally to a ‘spherical shape’ during their development in the *Lilium lancifolium* [7]. We found that a *P. ternata* bulbil emerges from a white punctate structure, which subsequently undergoes both longitudinal elongation and lateral expansion, ultimately developing into an oval spherical configuration. This progression is similar to the process of bulbil development in other species.

The initiation of bulbils started in the early plant growth stage, which does not require much nutrient accumulation from the maternal plant. Through three years of cultivation studies, we conclude that the phenotype (DB/SB) can be stably inherited, and all the experiment materials obtained by tissue propagation from a DB or SB maternal plant showed consistent phenotypes of DB or SB petioles, which is consistent with a previous report of the stable bulbil phenotype of *P. ternata* [12]. Therefore, bulbil initiation may be related to genetic/germplasm material. Our morphological and histological explorations suggest that the bulbil is differentiated from the parenchyma cells under the epidermis of the petiole, and the axillary bud could differentiate from the leaf axillary meristem, which supports the previous conclusions [13]. Until now, the regulatory mechanism underlying the formation and development of bulbils has remained incompletely understood. Available evidence suggests that the emergence of bulbils is intricately intertwined with environmental cues, phytohormonal regulation, and genetic factors [5,34,35].

### 3.2. Endogenous Hormones Regulate P. ternata Bulbil Initiation

Phytohormones, also known as plant hormones or growth regulators, regulate various processes in plant growth, development, and environmental adaptation at extremely low concentrations [7]. These processes work independently yet cooperatively to collectively regulate nutritional growth, reproductive growth, seed germination, seed maturation dormancy, embryo development, and adaptation to biotic and abiotic stresses [36,37,38]. In this study, we conclude that SA is the most critical compound for the initiation of bulbils and that it positively regulates the initiation; additionally, IPA, ABA, JA, and IAA are also important for bulbil regulation. There is a strong association between IPA and SA during bulbil initiation. During this stage, the IAA concentration decreases, whereas the ABA concentration increases. SA is an endogenous signal and regulates many physiological processes in plants. It provides protective effects against abiotic and biotic stresses, such as antioxidant responses, reducing light damage, and defense against drought stress, to protect plant cells from toxicity and death [39,40,41]. ABA is also involved in the plant’s response to stress. It modulates the stomatal aperture and regulates the expression of defense-related genes to resist environmental stresses [42]. Similarly, JAs are also essential plant hormones that regulate plant development and environmental adaptation, and they can resist bacterial infection and increase plant tolerance to abiotic stresses, such as ozone, ultraviolet light, high temperature, and freezing damage [43,44,45]. SA, ABA, and JA act alone or interact with each other or with other hormones among signaling pathways, which are indispensable in the defense against plant pathogens [46,47,48]. In this study, JA, ABA, and SA were highly enriched during bulbil initiation, showing the same patterns of change. Therefore, we speculated that bulbil production in *P. ternata* may also be an adaptive ability that gradually evolved during long-term adaptation to specific environmental phenomena during growth, such as global warming; drought or flood; and various bacteria, fungi, viruses, nematodes, and insects. This approach could help *P. ternata* propagate and withstand adverse conditions, and is not uncommon in many species [49,50].

Duan et al. have established a system for culturing cuttings derived from the hydroponic growth of *P. ternata*. In this system, they induced callus formation on the cut surfaces of leafy stems. Without the use of exogenous hormones, these calluses further developed and matured into tubers. To gain a deeper understanding of the biological mechanisms underlying callus formation, the researchers monitored changes in endogenous hormone levels. Their findings revealed significant alterations in auxin content during the initial 8 days of callus development. Specifically, the level of IAA decreased from the time of cutting until callus formation. Similarly, the contents of JA and jasmonic acid isoleucine also exhibited marked decreases. Notably, ABA content peaked on the 8th day, subsequently returning to its baseline level [1]. Based on the above results, we conclude that IAA and ABA exhibited a similar trend in the early stages of bulbil formation and callus formation, whereas JA demonstrated an opposing trend in change. We speculate that JA, as a stress response hormone, has a different metabolic rate and accumulation trend in plants under stress caused by physical injury vs. non-physical injury.

As an important cytokinin, IPA promoted the initiation of bulbil formation in *Lilium lancifolium* [51]. Similarly, we found that IPA was the second most important cytokinin, after SA, for the initiation of bulbils in *P. ternata*. In contrast, zeatin, another CTKs, was reduced during bulbil initiation. IPA is an iP-type CTK and zeatin is a tZ-type CTK, and the two have distinct tissue distribution patterns and convey diverse biological information [52]. We speculate that iP-type CTK might promote bulbil initiation and tZ-type CTK might be inhibited upon bulbil initiation in *P. ternata*.

SLs are important rhizosphere signaling molecules and a class of phytohormones that control shoot architecture [53]. As branching inhibitors, they inhibit tiller bud growth in rice. 5-DS is the SL with the simplest structure [54]. Brassinolide (BR) plays a crucial role in regulating various biological processes, such as cell division and expansion, stem elongation, root growth, shoot growth, and xylem differentiation [55]. BR can promote starch synthesis in the bulbil of *P. ternata*, thereby enhancing its expansion [3]. On the contrary, development of the bulbil was inhibited and SL content was increased with the application of propiconazole (a BR biosynthesis inhibitor, Pcz) [3]. In this study, the 5-DS content at the DU was significantly reduced by 28.5% compared to the SU. We speculate that the bulbil initiation mechanism is similar to that for the AM, and that SL is closely related to the initiation of *P. ternata* bulbils; that is, the content of SL in the AM during the bulbil initiation period was lower than that in the non-initiation period.

### 3.3. DEGs Involved in Phytohormone Biosynthesis and Signaling

The principal genes involved in hormone synthesis and signal transduction exhibit a profound correlation with the degree of hormone metabolism. Through our analysis, we have identified distinct genes that are associated with 5-DS, CTK, JAs, ABA, IAA, GA, and SA, and these genes display specific regulatory mechanisms among themselves.

In the 5-DS synthesis and signaling pathway, *D27 is* a regulatory gene upstream of the pathway [56], and *SMAX* exerts a negative regulatory effect on the biosynthesis of 5-DS [57]. In *Arabidopsis*, overexpression of *AtD27*-like1 in the d27 mutant effectively reinstated the phenotype of increased branching, thereby confirming the involvement of *D27* in the biosynthesis of SL [58]. In a chickpea and lentil study, the majority of *SMAX* and partner genes were found to exhibit a positive association with the branching level of plants [59]. In this study, the expression of *D27* was twofold lower in the DUs than in the SUs. Moreover, the expression of *SMAX* in the DUs was higher than in the SUs. The expression of these two genes corresponds to a similar decrease in 5-DS content, which is consistent with previously reported results. We speculate that *D27* and *SMAX* may be key regulatory genes for bulbil differentiation in *P. ternata*.

Nearly all living organisms produce CTKs. The initiation step of isoprenoid cytokinin biosynthesis is catalyzed with IPT [60]. CRE1 belongs to the category of histidine kinases (HKs). Cytokinin functions as a signaling molecule that triggers the signal input domain of HKs, leading to the autophosphorylation of the conserved histidine residue within the kinase domain [61]. B-type ARRs are positive regulators of cytokinin signal response. Type-A RRs are directly regulated by type-B RRs [60]. An examination of floral bud development in *Alfalfa* revealed that during the biosynthesis of CTK, a noteworthy upregulation of one *IPT* gene and two *CYP735A* genes was observed in the samples exhibiting elevated CTK concentrations. Furthermore, within the cytokinin signaling pathway, distinct expression patterns were discerned among two *AHP*, four *B-ARR*, and five *A-ARR* genes in the samples containing high levels of CTKs [62]. In the present study, we identified all of the *CRE1*, *AHP*, *A-ARR*, and *B-ARR* unigenes expression levels were up-regulated during the bulbil initiation period, and were positively correlated with IPA content.

In JA biosynthesis, phospholipases release α-linolenic acid, which is then catalyzed with LOX to form 13-hydroperoxy-9,11,15-octadecatrienoic acid as an intermediate. AOS and AOC then modify it to produce 12-oxo-phytodienoic acid, a direct precursor of JA. JA is synthesized in peroxisomes through OPR3 [63]. This reduction step is critical for the subsequent steps in JA biosynthesis. JA is converted into MeJA by JMT [64]. In the JA signaling pathway, the receptor COI1 receives jasmonate, an F-box protein interacting with ASK1 and ASK2. JAZ repressors negatively regulate jasmonate signaling by suppressing transcription factors. MYC2, a bHLH transcription factor, is a JAZ-binding factor that targets jasmonate-responsive genes and regulates jasmonate-mediated processes [65]. The key genes in the JA biosynthetic pathway were screened out from hormonal treatments and biotic stress-related barley transcriptome datasets, and *AOS*, *OPR3*, and *LOX2* exhibited a significant positive correlation and interaction among themselves [66]. In this study, transcriptome sequencing showed that the expression levels of *AOS*, *AOC*, *OPR*, and *JMT* had the same pattern of expression, and *JAZ* and its binding factor *MYC2* were significantly upregulated during bulbil initiation.

ABA is synthesized from β-carotene via the oxidative cleavage of neoxanthin and conversion of xanthoxin to ABA via ABA-aldehyde. The abiotic stress-induced activation of many ABA biosynthetic genes, such as *ZEP*, *NCED*, *AAO*, and *MCSU*, appear to be regulated through calcium-dependent phosphorylation pathways [67]. *PP2C* negatively regulates ABA-induced gene expression, and *SnRK2* is a critical positive regulator of ABA signaling. Following the ABA treatment of *Panax notoginseng* seeds, the expressions of *PYL* and *SnRK2s* increased, whereas the expression of *PP2C* decreased. The expression of *PP2C* was positively correlated with the content of ABA [68]. In this study, we conclude that during *P. ternata* bulbil initiation, the ABA content increased, *PP2C* expression elevated, and *SnRK2* expression decreased. This is consistent with the conclusions of previous studies.

In the IAA biosynthesis pathways, the indole-3-pyruvic acid (IPyA) pathway has been established as the prevailing route in plants. IPyA is decarboxylated to IAA in a rate-limiting and irreversible reaction catalyzed with B flavin-containing monooxygenases from the YUCCA family [69]. In the first step of the tryptamine biosynthesis pathway of IAA, TDC catalyzes tryptophan to tryptamine [70]. ALDH encodes the enzyme that catalyzes indole-3-acetaldehyde to IAA [71]. The transduction of the IAA signal depends on three protein families: Aux/IAA, TIR1/AFB, and ARF. High IAA concentration accelerates protein hydrolysis, relieving suppression of the auxin response genes by facilitating ARF dimer formation. The TIR1/ABF protein interacts with Aux/IAA, leading to its polyubiquitination and degradation by the proteasome. This pathway efficiently activates auxin response genes, including Aux/IAA and the GH3 family, triggering a negative feedback loop. Conversely, low IAA levels cause *Aux/IAA* inhibitors to suppress *ARF* activity [72]. In Lily, wounding leads to an interruption of auxin transport, thus inducing bulbils. Additionally, *AUX/IAA*, *ARF*, and *GH3* were highly expressed during bulbil formation [73,74]. In this study, we conclude that *AUX/IAA* was downregulated while the expressions of *AUX1*, *GH3*, and two *ARF* were upregulated in the DUs compared to the SUs. The auxin signaling pathway, explaining most auxin transcriptional responses, relies on a de-repression mechanism. Auxin degrades Aux/IAAs, releases ARFs, and activates transcription. However, this model is not suitable for all ARFs. In Arabidopsis, only 5 of 22 ARFs fit the model, as they interact with Aux/IAAs. The remaining 17 have limited repressor interaction, and their mechanisms are unclear [75]. The mechanisms of *AUX/IAA* and *ARF* in bulbil initiation require further study.

GA biosynthesis is a complicated pathway that is mainly catalyzed by seven key steps: CPS, KS, KO, KAO, GA13ox, GA20ox, and *GA3ox* [76]. GA20ox serves as the rate-limiting enzyme in the synthesis of GA [77]. The deactivation of bioactive GAs is controlled by several distinct groups of enzymes, among which GA2ox plays a pivotal role [76]. This enzyme catalyzes the inactivation of GA or its precursors, thereby enhancing plant architecture and mitigating recalcitrance in biomass production [78]. Conversely, the DELLA protein functions as a signaling inhibitor of GA, regulating its activity within the plant [79]. In the GA synthesis pathway, lower *GA20ox* expression and greater *GA2ox* expression were detected in the DUs compared to the SUs. In this study, *GA2ox* was negatively correlated with *DELLA* expression, which is consistent with the regulatory relationship reported above.

There are two SA synthesis pathways. One pathway begins with phenylalanine and converts it to t-cinnamic acid and benzoic acid. The enzyme PAL, which converts phenylalanine to t-cinnamic acid, is crucial for the production of phenylpropanoid compounds. The other pathway starts with chorismate and proceeds through isochorismate, with ICS being the rate-limiting enzyme [80]. In the SA signaling pathway, SA binds to its receptor NPR1. In turn, *NPR1* interacts with the TGA transcription factors. This interaction serves as a transcriptional co-activator, leading to the induced expression of *PR* genes [81]. In the synthesis of SA within the regulatory framework that governs bulbil of *Lycoris radiata* initiation, the enzymes ICS and PAL, which play pivotal roles, have exhibited upregulated expression levels [82]. *TGA*, which is involved in Lily defense, was highly expressed during bulbil formation [74]. In this study, it was also found that *PAL*, *NPR*, *PR-1* and *TGA* were upregulated during bulbil initiation in *P. ternata.*

### 3.4. DEGs Involved in the AM Initiation Regulatory Network

Utilizing a comprehensive understanding of cellular and tissue development mechanisms, scientists have discovered that the formation of plant bulbils is attributed to the meticulous regulation of gene expression within the meristematic tissue cells of plants. This regulation ultimately results in the attainment of totipotency in plant cells [83]. AM initiation and growth are controlled by a complex and interconnected regulatory network [84]. In *Arabidopsis*, STM has plays a crucial role in establishing and maintaining the apical and AMs [19]. STM was shown to upregulate the expression of CTK biosynthesis genes to activate WUS and promote the germination of AMs in the axils [85]. The WUS protein also activated STM expression and directly promoted CLV3 expression upon binding [86,87]. The process of AM initiation involves intricate protein movements, transcriptional regulation, protein–protein interactions, and feedback control mechanisms across various pathways. Central to this complex network is the *STM* gene, and its expression significantly impacts meristematic activity. *STM* was expressed at low levels in leaf axils, which was sufficient to maintain the meristematic capacity of cells [17,20]. *STM*, together with homologous genes in other species, including *OSH1* in rice and *knotted-1* in maize, belongs to the *KNOX* gene family [88,89]. The *BLH9* and *BLH8* TFs both promote normal meristematic structure by interacting with *STM* TFs in the *KNOX* family [90]. In this study, transcriptome and qRT-PCR validation showed that in *P. ternata*, *KNOX* gene expression was abnormally increased during bulbil initiation and was hardly present in other sites. Furthermore, we identified other DEGs involved in bulbil initiation in *P. ternata*, e.g., *WUS*, *CVL*, *ATH1*, *SRL*, *RAX*, and *LEAFY* that were associated with hormone biosynthesis and various signaling pathways. The same results were also seen in *P. ternata* bulbil initiation induced by phytohormones, and the key genes and pathways involved in bulbil development were mainly enriched in plant hormone signal transduction metabolism pathways. We confirmed that the expression patterns of some DEGs, such as *CRE1*, *A-ARR*, *B-ARR*, *AUX1*, *ARF*, *AUX/IAA*, *GH3*, *GA2ox*, *GA20ox*, and *IPT*, were closely related to hormone content. So, we speculate that bulbil initiation is regulated by a complex network that includes hormones-related genes and regulate AM key genes formation (Figure 9). We also obtained the same result by constructing a co-expression network (Appendix A, Appendix A). The genes involved in different hormone biosynthetic and signal transduction pathways and hormone contents showed a certain regularity. Combined with the annotation information in the results of this study, these genes can further screened in future research.

### 3.5. DEGs Involved in Sucrose and Starch Metabolic Pathways

Numerous studies have shown that bulbil formation is a multi-factor regulatory process involving hormones, glucose metabolism, and transcriptional regulatory networks [73]. Although this study focused on the effects of hormones and transcriptional regulation on the initiation of *P. ternata* bulbils, the transcriptome results show that the starch and sugar metabolic pathway genes were among the enriched DEGs. Previous research conducted by Guo et al. has documented the influence of BR on the development of *P. ternata* bulbils. BR treatment promoted starch accumulation by upregulating the expression of *TCH4* in the BR signal transduction pathway, while the inhibition of BR upregulated *PP2C* expression [91]. The comparative analysis of the DU and SU transcriptomes in this study demonstrated that *TCH4* was upregulated and *PP2C* was downregulated, which is consistent with the findings of Guo et al. Based on our transcriptome sequencing and KEGG analysis, we also identified 75 DEGs in the sucrose and starch metabolic pathways, among which 54 genes were upregulated. In this study, the expression of *TCH4* in DUs is less than one-fourth of that in SUs. Transcriptome data show that the expression of three *PP2C* genes (Cluster-8921.65947, Cluster-8921.50405, Cluster-8921.57559) were upregulated in DUs vs. SUs, while two *PP2C* (Cluster-8921.50654, Cluster-8921.46119) were downregulated. It is speculated that *TCH4* has little effect on the regulation of sucrose and starch metabolism in the beginning period of bulbil initiation, while *PP2C* may be related to sucrose and starch metabolism. 

### 3.6. DEGs Involved in the Trehalose-6-Phosphate-Triggered Signaling Pathway

Trehalose-6-Phosphate (T6P), serving as the precursor of trehalose, serves as a crucial signaling molecule that plays a pivotal role in regulating plant growth and development in response to carbon availability [92]. It forms trehalose through the promotion of trehalose 6-phosphate synthase, which causes condensation between UDP-glucose and glucose-6-phosphate [93]. Simultaneously, it is noteworthy that, rather than functioning as trehalose itself, T6P serves as a pivotal regulatory signal in the process of plant growth and development [94]. It can be used as a regulator of carbon metabolism and an enhancer of photosynthetic capacity [95]. Research indicates that in plants, T6P functions as a signaling molecule capable of facilitating hexokinase (HXK) in sensing carbon status, akin to its role in yeast. The presence of sucrose-induced T6P is conducive to enhancing plant growth. Conversely, the absence of exogenous carbon sources results in T6P suppressing growth, suggesting that T6P, as a signaling molecule, coordinates environmental conditions with various other signaling cascades to regulate plant growth and development [96]. More and more studies demonstrated that T6P serves as a crucial signaling molecule that bridges plant metabolism with growth and development [97]. In *Arabidopsis* seedlings, T6P plays an important role in signaling below micromolar concentrations and participates in the integration of carbon and energy states to regulate plant growth and development [96]. T6P also can indirectly induce redox reactions, activate AGPase to promote starch synthesis, and inhibit the activity of *SnRK1* [98]. T6P regulates root branching via the essential kinases *SnRK1* and Target of *TOR*, with auxin, a plant hormone, playing a pivotal role. Elevation of T6P levels, achieved through genetic targeting in lateral root founder cells as well as light-induced liberation of the presignaling T6P-precursor, demonstrates that T6P promotes root branching by synergistically suppressing *SnRK1* and stimulating *TOR* [99]. The expression of a heterologous *TPP* to decrease *Tre6P* levels in axillary buds results in a substantial retardation of bud outgrowth under long-day conditions and suppression of branching in short-day environments [100]. In addition to its role in plants, T6P may exert regulatory functions in embryonic development and other biological processes by modulating the quantity of sugar diverted towards glycolysis and maintaining metabolic homeostasis. TPS1 oversees the *Arabidopsis* response to sugar concentrations, thereby collaborating with carbohydrates to influence plant metabolism and development, and assumes a pivotal role in embryonic maturation [101]. The level of T6P increased significantly when sucrose was supplied or light was restored after darkness [96]. Increasing or decreasing the level of T6P can affect the response of *Arabidopsis thaliana* to sugar, indicating that T6P can affect plant growth and carbon utilization [96]. In addition, studies have confirmed that T6P content is also closely related to wheat grain development [102]. These results suggest that T6P can reflect the availability of sucrose-form carbon in plants and regulate plant growth [103]. The T6P pathway, operating independently of the photoperiod pathway, is capable of exerting influence on the expression of flowering and flowering pattern genes within the age pathway in the apical meristem of plants. This process serves to furnish the bodily signal that is dictated by the development of the meristem, specifically under conditions of carbon supply [104]. Therefore, T6P acts as a signal to coordinate floral induction by controlling the expression of key flowering integration genes in leaves and apical meristems [105]. These results indicate that T6P, as a signal, can sense the level of carbohydrates in plants and regulate the flowering of plants together with other signals. In the present study, transcriptome data analysis revealed the presence of three *TPS* and one *HXK*, genes that exhibited differential expression patterns in DU vs. SU. Among them, two of these genes (Cluster-8921.44959 and Cluster-8921.58367) were observed to be highly expressed, whereas the third gene (Cluster-8921.5138) demonstrated low expression level, and the *HXK* was downregulated in DU. While the specific mechanism underlying the influence of T6P on bulbil development remains to be thoroughly explored and elucidated.

### 3.7. Plant–Pathogen Interactions Regulating Bulbil Initiation

Based on our analysis of the KEGG pathway, it was discovered that the plant–pathogen interactions pathway exhibited the highest enrichment of DEGs. Plant–pathogen interactions induce a signal transmission series that stimulates the plant’s host defense system against pathogens and this, in turn, leads to disease resistance responses [106]. There is extensive signal exchange and recognition in the process of triggering the plant immune signaling network. Plant messenger signaling molecules, such as calcium ions, reactive oxygen species, and nitric oxide, and plant hormone signaling molecules, such as SA and JA, play key roles in inducing plant defense responses [107]. In this study, SA was found to be the most influential hormone on the initiation of bulbil among the 12 measured hormones. In recent years, research on the function of SA has become an important and rapidly developing field in biology. SA plays a wide range of physiological roles in plant growth, development, maturation, aging regulation, and stress-resistance induction. However, the current research on the physiological roles of SA in plants is still focused on disease resistance and signal transduction [108,109,110]. After infection, SA is closely related to the formation of the local resistance of infected tissues and the SAR of uninfected tissues. Infected tissues show the HR response, produce signaling substances such as SA, and activate *SAR* gene expression [111,112]. SA, as a system signal, can also cause an increase in the SA level in uninfected tissues and then induce the expression of *PR* genes, resulting in disease resistance throughout the whole plant [113]. JA is also a stress signal molecule that accumulates rapidly and massively when plant tissues are invaded by microbial pathogens or insects [114]. JAs may induce the expression of defense-related proteins, such as polyphenol oxidase, protease inhibitors, peroxidase, and lipoxygenase. JAs may also induce the production of alkaloids and some volatiles, and the formation of defense structures, which exert the stress and disease resistance functions of plants [115]. Therefore, we hypothesize that DEGs on the plant–pathogen interactions pathway were related to the changes of hormones, especially SA and JA. The formation of bulbil may be the result of evolution after long-term environmental stress.

### 3.8. Major Genes and Transcription Factors Involved in Regulating the Phenotype

TFs play a crucial role in plant morphogenesis, as they act as a magical key that unlocks the multiple codes of plant development and growth. In complex biological networks, TFs regulate the expression patterns of genes by binding to DNA, thus shaping the unique morphology and structure of plants. In different stages of plant morphogenesis, TFs plays different functions [116]. In yam, the Aux/IAA, E2F, MYB, and bHLH families have been shown to play key roles in triggering bulbil formation [117]. 

In this comprehensive analysis, we discovered ERF, bHLH, MYB, E3, NAC, WRKY, GRAS, etc. TFS (Appendix A) exhibit different expression patterns in *P. ternata* bulbil initiation. Notably, KNOX, a transcription factor responsible for regulating morphogenesis and hormone metabolism, demonstrated an elevated expression level during the initial stages of bulbil formation, thereby serving as a pivotal regulatory factor in the bulbil initiation process. With further study of the molecular biology, we hope that more TF mechanisms of action will be revealed.

### 3.9. MAPK Pathway Involved in Regulating the Phenotype

At present, numerous protein kinases have been identified in plants, with MAPK being the largest and most crucial class among them. MAPK serves as the pivotal signaling molecule in plants, operating downstream of sensors or receptors to orchestrate cellular responses effectively [118]. It is a highly conserved biological signal transduction module that is widely present in eukaryotes and plays a crucial role in plant growth and development and resistance to stress [119]. MAPK mediating a variety of external signals to play a role in appropriate cellular responses, and MAPK pathway have a wider range of stimuli in plants [120]. Plant hormones are important signaling molecules whose biosynthesis or transport often occurs when cells are stimulated by external stimuli. Plant perception of internal and external stimuli often triggers rapid activation of the MAPK cascade in a short period of time [121]. This rapid response allows it to regulate the biosynthesis or hormone transport. The biosynthesis, transport, and signaling of plant hormones are intricately related to MAPK signals, and some MAPKs members act as upstream regulators to control hormone biosynthesis or transport [122]. However, other MAPKs are located downstream and regulate hormone signaling, and *MAPK*s are also activated by SA, JA, and ETH. In *Arabidopsis thaliana*, MAPKs are involved in the synthesis of JA, ethylene, and SA, as well as the signal transduction of ethylene, SA, ABA, and JA [123,124]. Studies have shown that JA regulates MAPK activity and *MAPK* gene expression. Studies have shown that MeJA can enhance the resistance of plants to abiotic stress by increasing the activity of antioxidant enzymes [125]. MeJA strongly induced 14 *CmMAPKs* in melon [126]. Additional investigation revealed that the enzymatic activities of catalase and peroxidase in both transgenic and wild-type *Populustrichocarpa* plants exhibited marked elevations at the 12 h mark following exposure to MeJA. Notably, the magnitude of this increase was more pronounced in the overexpressed plants compared to their wild-type counterparts [127]. After 6 h of SA treatment, the transcription level of *ShMAPK5* in *Saccharumofficinarum* was upregulated [128]. SA and JA can significantly induce the expression of SmMAPK3 in *Salvia miltiorrhiza* [129].

In the present investigation, a noteworthy enrichment of DEGs was observed within the MAPK signaling pathway when comparing DU to SU. Notably, seven *MAPK* genes exhibiting distinct expression patterns were identified. Specifically, the expression levels of Cluster-8921.49233, Cluster-8921.77773, Cluster-8921.59971, and Cluster-8921.65755 were upregulated, whereas the expression leves of Cluster-8921.52183, Cluster-8921.46530, and Cluster-8921.60236 were downregulated. Based on these findings, we hypothesize that MAPK and its signal transduction mechanisms, in conjunction with hormonal regulation, play a pivotal role in governing the developmental processes of the bulbil.

## 4. Materials and Methods

### 4.1. Source and Identification of Plant Materials

In this study, the bulbils that grew at both the top and base of the *P. ternata* petiole were referred to as DB. The bulbils that grew only at the base of the petiole were designated SB (Figure 10). The SB and DB samples were collected in Nanchong, Sichuan (longitude: 106°33′43″, latitude: 31°04′44.15″), in May 2018. They were propagated and grown for three years in a growth chamber at Sichuan Agricultural University, Chengdu. Both the phenotypes showed stable inheritance. In the third year, underground corms were dug, cleaned, and stored at 4 °C.

### 4.2. Morphology of the Top Bulbil in Double-Bubil P. ternata

Tissue samples were collected from the top petiole of the DB during three distinct stages of bulbil development, i.e., bulbil formation (BF), bulbil expansion (BE), and bulbil maturation (BM). After vacuum freeze-drying, the samples were scanned using multiangle computed tomography (CT) with an Xradia 510 Versa 3D X-ray microscope (Carl Zeiss Microscopy GmbH, Gottingen, Germany). The reconstructed 2D slices were then observed using ZEISS’s 3D Viewer software (ZEISS Scout and scan V16). Three-dimensional visualization software was used to generate three-dimensional-rendering color images for each virtual slice of the tissue samples.

Healthy tubers from the DB and SB of *P. ternata* were cultivated in an illuminated incubator (ILI) with optimal light conditions. Post-germination, the morphology of the top petiole was continuously monitored and documented. Samples were fixed in a 70:5:5 ethanol/formalin/acetic acid (FAA) solution for 24 h at 4 °C. Subsequently, dehydration was performed through a graded ethanol series, followed by paraffin embedding. Longitudinal sections (10 μm) were prepared using a rotary microtome (RM) and stained with safranin and alcian green, adhering to standard histological staining protocols (SHSP). The stained sections were observed and documented using a digital pathology section scanner (Pannoramic MIDI, 3DHISTECH Ltd. Budapest, Hungary).

### 4.3. Sample Preparation for Transcriptomic and Targeted Metabolomic Analyses

The tissue-cultured tubers were induced from tender petioles of the DB and SB, and they were transplanted into nutrient soil after rooting. After two growth cycles, *P. ternata* with trifoliate compound leaves were ready for sampling.

Samples were taken from the DB and SB during the same growth period, named as shown in Figure 4A: SM (SB petiole without bulbil), SU (top SB petiole without bulbil), DM (DB petiole without bulbil), and DU (top DB petiole with tender bulbil). The samples were washed with distilled water, frozen in liquid nitrogen, and stored at −80 °C. Each sample was from at least 30 plants, with three biological replicates for each tissue sample. The DU, SU, DM, and SM were used for transcriptomic analysis, while the DU and SU were used for metabolomic analysis.

### 4.4. Targeted Metabolomics for Plant Hormone Detection

The plant endogenous hormones IAA, zeatin, isopentenyladenine (IP), IPA, indole-3-butyric acid (IBA), trans-zeatin nucleoside (TZR), SA, methyl salicylate (MESA), JA, MeJA, ABA, and 5-DS were quantified using the electrospray ionization (ESI)-HPLC–MS/MS method at Shanghai Biotree Biotech Co., Ltd. (Shanghai, China) The analysis was performed on an Agilent 1290 HPLC (Agilent, Santa Clara, CA, USA)) tandem AB Corporation Qtrap6500 mass spectrometer (SCIEX, Marsiling, Singapore). The DU and SU samples were ground using liquid nitrogen, and three biological replicates of each sample were included in the experiment. Except for 5-DS, the other conventional acid–alkaline hormones were tested together.

### 4.5. Detecting Conventional Acid–Alkaline Hormones

Samples were pulverized in liquid nitrogen, weighed, and transferred to EP tubes. A 10 mL volume of isopropyl alcohol/hydrochloric acid buffer was added, and the tubes were shaken for 30 min at 4 °C. Then, 20 mL of dichloromethane was added and the tubes were vortexed for 30 min at 4 °C. The tubes were centrifuged at 4 °C, 13,000 r/min, for 5 min. The supernatant was removed, and the lower organic phase was blow-dried with nitrogen and dissolved in 400 µL of MeOH with 0.1% formic acid. The solution was filtered through a 0.22 μm filter and placed in a fresh tube for LC-MS/MS analysis.

In HPLC, a poroshell 120 SB-C18 (Agilent, Santa Clara, CA, USA) reverse-phase chromatography column (2.1 × 150, 2.7 μm) was used, with a column temperature of 30 °C; the mobile phase was A:B = (methanol/0.1% formic acid): water/0.1% formic acid. The elution gradient was 0–1 min, A = 20%; at 1–9 min, A increased to 80%; at 9–10 min, A = 80%; at 10–10.1 min, A decreased to 20%; and at 10.1–15 min, A = 20%. The injection volume was 2 µL. The mass spectrometry conditions were as follows: air curtain gas, 15 psi; spray voltage, 4500 V; nebulization gas pressure, 65 psi; auxiliary air pressure, 70 psi; and nebulization temperature, 400 °C. The selected reaction monitoring conditions for protonated or deprotonated plant hormones ([M+H]^+^ or [M−H]^−^) are detailed in Appendix A.

### 4.6. Detection Method for 5-Deoxystrigol

Samples were pulverized in liquid nitrogen, weighed, and transferred to EP tubes; 5 mL of ultrapure water was added, and the mixture was left to stand for 10 min. Then, 10 mL of acetonitrile was added and the mixture was heated in a water bath for 15 min. Then, the tubes were stored overnight at −20 °C. The tubes were centrifuged at 8000 rpm for 5 min, and the supernatant was collected. The organic phase was blow-dried with nitrogen and dissolved in 3 mL of n-hexane. Then, it was eluted through an Agilent FL pillar (Agilent, Santa Clara, CA, USA) (500 mg, 6 mL) with 5 mL of hexane, and the organic phase was blow-dried with nitrogen and dissolved in 200 µL of methanol. The supernatant was filtered through a 0.22 µm filter and placed in a fresh tube for LC–MS/MS analysis.

In HPLC, a poroshell 120 SB-C18 reverse-phase chromatography column (2.1 × 150, 2.7 µm) was used, with a column temperature of 30 °C; the mobile phase was A:B = (methanol/0.1% formic acid):water/0.1% formic acid. The elution gradient was 0–5 min, A = 80%; the injection volume was 2 µL. The mass spectrometry conditions were as follows: air curtain gas, 25 psi; spray voltage, 5000 V; nebulization gas pressure, 55 psi; auxiliary air pressure, 65 psi; and nebulization temperature, 350 °C. The selected reaction monitoring conditions for protonated 5-DS ([M+H]^+^) are detailed in Appendix A.

### 4.7. Metabolome Data Analysis

PCA (principal component analysis) [130] and OPLS-DA (orthogonal projections to latent structures discriminant analysis) [131] were performed using SIMCA software (V16.0.2, Sartorius Stedim Data Analytics AB, Umea, Sweden). The OPLS-DA permutation test randomly changed the order of the categorical variable, Y, multiple times to construct the corresponding OPLS-DA model, thus obtaining the R^2^ and Q^2^ values of the stochastic model. The differentially abundant metabolites were identified based on the following criteria: VIP (Variable Importance in the Projection) > 1 and *p* < 0.05.

### 4.8. Transcriptome Sequencing and Bioinformatics Analysis

The RNA Nano 6000 Assay Kit and the Bioanalyzer 2100 (Agilent, Santa Clara, CA, USA) were used to determine the RNA’s concentration and integrity. mRNA was purified from total RNA using poly-T oligo-attached magnetic beads. Fragmentation was performed in first-strand synthesis reaction buffer with divalent cations and a high temperature. cDNA synthesis was performed using random hexamer primers, M-MuLV Reverse Transcriptase, RNaseH, DNA Polymerase I, and dNTP. Overhangs were converted to blunt ends with exonuclease/polymerase activities. Adaptors were ligated to DNA fragments after 3′ end adenylation. The library fragments were purified with the AMPure XP system, and PCR amplification was performed. Libraries were pooled based on concentration and target data amount, and then sequenced on an Illumina NovaSeq 6000 (Agilent, Santa Clara, CA, USA).

All the downstream analyses were performed on high-quality, cleaned data. The clean reads were assembled using Trinity (v2.6.6) [132] software using the reference sequence. Then, the FPKM (Fragments Per Kilobase of transcript sequence per Millions base pairs sequenced) conversion of the Readcount number was performed [133].

### 4.9. Analysis of Differentially Expressed Genes (DEGs)

The differential expression genes (DEGs) between the SU and DU groups were analyzed using the DESeq2 R package version 1.20.0. DESeq2 employed a model based on the negative binomial distribution to accurately identify differential expression patterns from digital gene expression data [134]. To ensure the reliability of our findings, the resulting *p*-values were adjusted using the Benjamini and Hochberg method to control the false discovery rate. Significant differential expression was determined using the following thresholds: padj < 0.05 and |log2(fold change)| > 1. GOseq 1.10.0 and KOBAS v2.0.12 software were employed for Gene Ontology (GO) functional enrichment analysis [135] and Kyoto Encyclopedia of Genes and Genomes (KEGG) [136] pathway enrichment analysis of the DEGs, respectively. These analyses were conducted based on the hypergeometric distribution principle. The background gene sets encompassed all genes exhibiting no significant differences in expression and those not annotated in the GO and KEGG databases.

### 4.10. Verification of Gene Expression Using Qualitative Real-Time Polymerase Chain Reaction (qRT–PCR)

The DEGs identified from the transcriptome sequencing were validated using qRT–PCR. Primer 3 software (https://primer3.ut.ee/ (accessed on 9 March 2024)) was used to design qRT–PCR primers for the selected genes; their sequences are shown in Appendix A. Three biological replicates were used for each selected gene. RNA extraction was performed using the TRIzol polysaccharide polyphenol RNA kit (Invitrogen, Waltham, MA, USA), and cDNA was synthesized using reverse transcriptase (Invitrogen) according to the kit’s protocol. qRT–PCR was conducted with a Bio-Rad qPCR machine using the SYBR Green I Master Mix (TaKaRa, Dalian, China). The 20 µL qPCR mixture included 5 µL of SYBR qPCR master mix (Vazyme, Nanjing, China), 0.4 µL of each primer (10 µM), 2.5 µL of cDNA, and 11.7 µL of RNase-free water. The expression levels were normalized to those of the reference gene, *P. ternata tubulin* [137], and the relative expression levels of the genes were calculated using the 2^−ΔΔCT^ method. We utilized the Pearson correlation coefficient to assess the correlation between real-time RT-PCR values (2^−deltaCt^) and FPKM values derived from RNA-seq for each gene. A Pearson correlation threshold of 0.8 and a *p*-value threshold of 0.05 were employed to ensure the accuracy and reliability of the analysis.

## 5. Conclusions

In this study, we found that the crucial period for top bulbil initiation in *P. ternata* occurs prior to complete leaf spread, and transcriptomic and targeted metabolomic analyses were performed in the bulbil initiation stage to unearth the intricate relationship between hormones, genes, and bulbil development. Twelve phytohormones were quantified, from which eight differentially expressed hormones were identified; JA, ABA, IPA, and SA were highly enriched, while the IAA, zeatin, MeJA, and 5-DS contents were decreased. SA is the most critical for the initiation of bulbils and positively regulates the initiation. IPA, ABA, JA, and IAA are also important for bulbil regulation. Moreover, there is a strong association between IPA and SA. The transcriptional changes in *IPT*, *CRE1*, *A-ARR*, *B-ARR*, *AUX1*, *ARF*, *AUX/IAA*, *GH3*, *SAUR*, *GA2ox*, *GA20o*x, *AOS*, *AOC*, *OPR*, *JMT*, *COI1*, *JAZ*, *MYC2*, *D27*, *SMAX*, *PAL*, *ICS*, *NPR1*, *TGA*, *PR-1*, *MCSU*, *PP2C*, and S*nRK* were highly correlated with hormone concentrations, indicating that bulbil initiation is coordinately controlled by multiple phytohormones. Notably, eight TFs that regulate AM initiation have been identified as pivotal regulators of bulbil formation. Among these, *WUS*, *CLV*, *ATH1*, and *RAX* have been observed to exhibit elevated expression levels. Conversely, *LEAFY* demonstrated contrasting expression patterns. The intricate expression profiles of these TFs are closely associated with the upregulated expression of *KNOX*, suggesting a intricate regulatory network underlying the complex process of bulbil initiation. This research provides valuable insights into bulbil development and could be conducive to optimizing molecular breeding strategies targeting *P. ternata.* In the future, we will further study the function of *KNOX* in hormone-induced bulbil development.

## Figures and Tables

**Figure 1 ijms-25-06149-f001:**
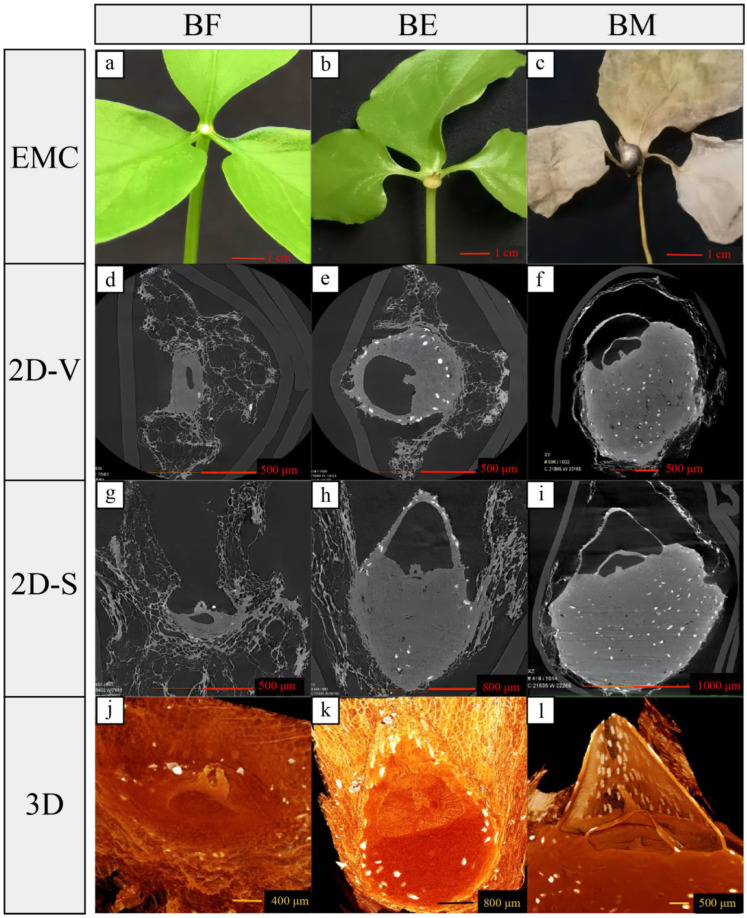
Morphological characteristics of the top bulbil in different periods of development. Each column shows the same period of development for the top bulbil. BF, bulbil formation; BE, bulbil expansion; BM, bulbil maturity; EMC, external morphological characteristic of bulbil; 2D-V, 2D images of the bulbil transverse section; 2D-S, 2D image of bulbil longitudinal section; 3D, three-dimensional rendering image. EMC of BF, BE, and BM (**a**–**c**); 2D-V of BF, BE, and BM (**d**–**f**); 2D-S of BF, BE, and BM (**g**–**i**); 3D of BF, BE, and BM (**j**–**l**).

**Figure 2 ijms-25-06149-f002:**
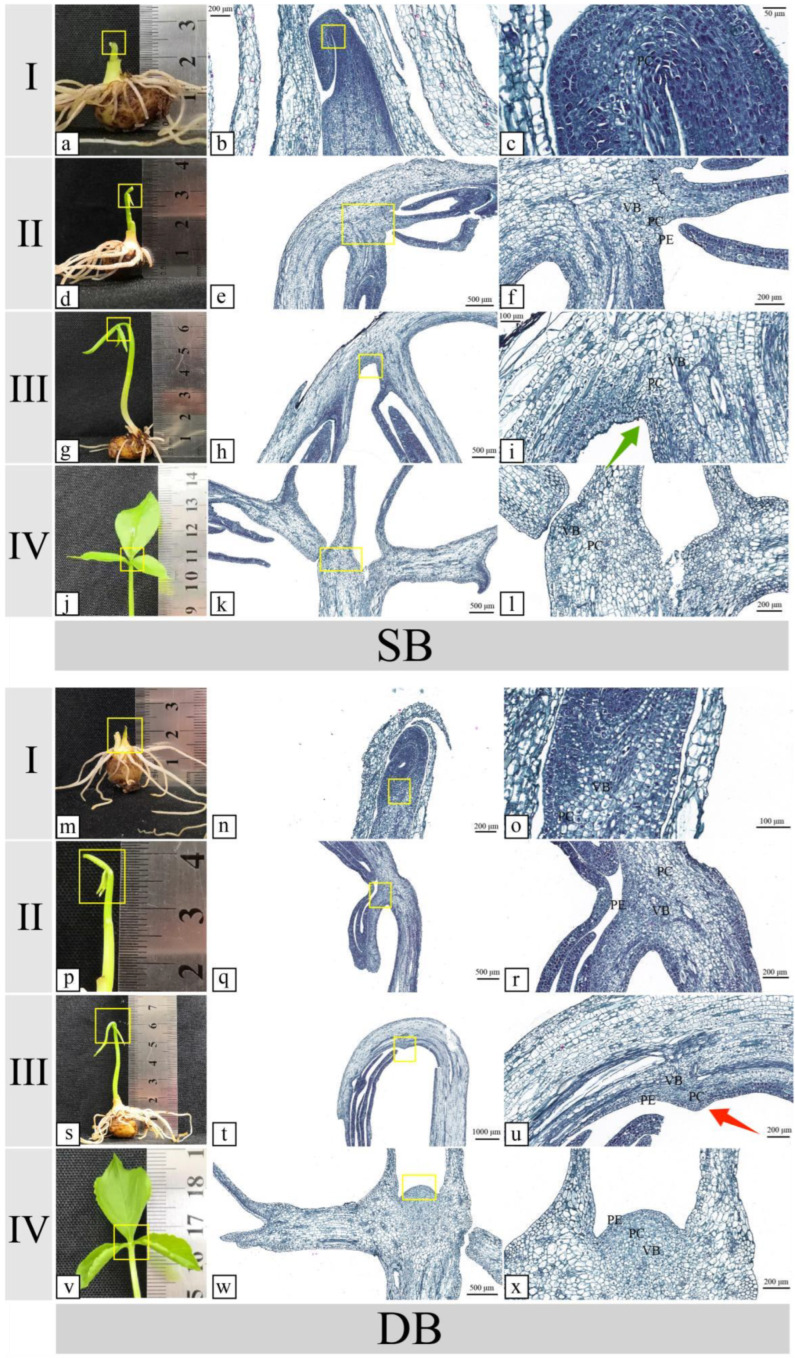
Paraffin section pictures of SB and DB petioles at different growth stages of *P. ternata*. SB, single-bulbil type; DB, double-bulbil type; VB, vascular bundle; PC, parenchyma cell; PE, petiole epidermal cell. I, II, III, and IV represent the different development stages of SB and DB from germination to leaf spreading. The (**a**,**d**,**g**,**h**) morphological pictures of the top of the petiole and leaves of SB petioles in I, II, III, and IV stages, respectively; the (**m**,**p**,**s**,**v**) morphological pictures of the top of the petiole and leaves of DB petioles in I, II, III, and IV stages, respectively; (**b**,**e**,**h**,**k**,**n**,**q**,**t**,**w**) correspond to the paraffin section diagrams of the yellow-boxed areas within (**a**,**d**,**g**,**j**,**m**,**p**,**s**,**v**), respectively; (**c**,**f**,**i**,**l**,**o**,**r**,**u**,**x**) correspond to enlarged images of the yellow-boxed areas within (**b**,**e**,**h**,**k**,**n**,**q**,**t**,**w**), respectively. The red arrow shows where the top bulbil germinated in DB, while the green arrow indicates where no parenchymal hyperplasia occurred in SB during the same time.

**Figure 3 ijms-25-06149-f003:**
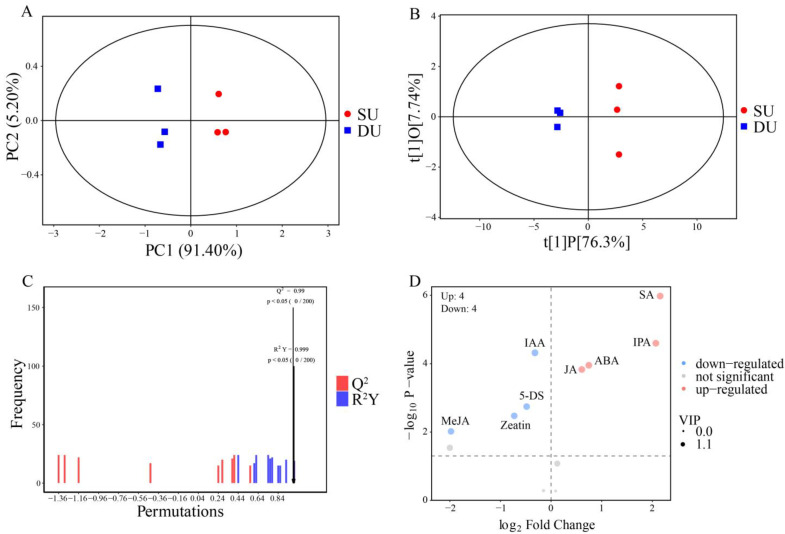
Differentially accumulated metabolites in DUs and SUs. (**A**) PCA of the metabolites identified in DUs and SUs. (**B**) OPLS-DA plots and loading plots of DU vs. SU. (**C**) Permutation test of the OPLS-DA model for comparing SU and DU. (**D**) Volcano plot for DU vs. SU. SA, salicylic acid; IPA, isopentenyl adenosine; ABA, abscisic acid; JA, jasmonic acid; IAA, indole-3-acetic acid, 5-DS, 5-dexoxystrigol; MeJA, methyl jasmonate; SU, the top of the petiole in SB; DU, the top of the petiole in DB.

**Figure 4 ijms-25-06149-f004:**
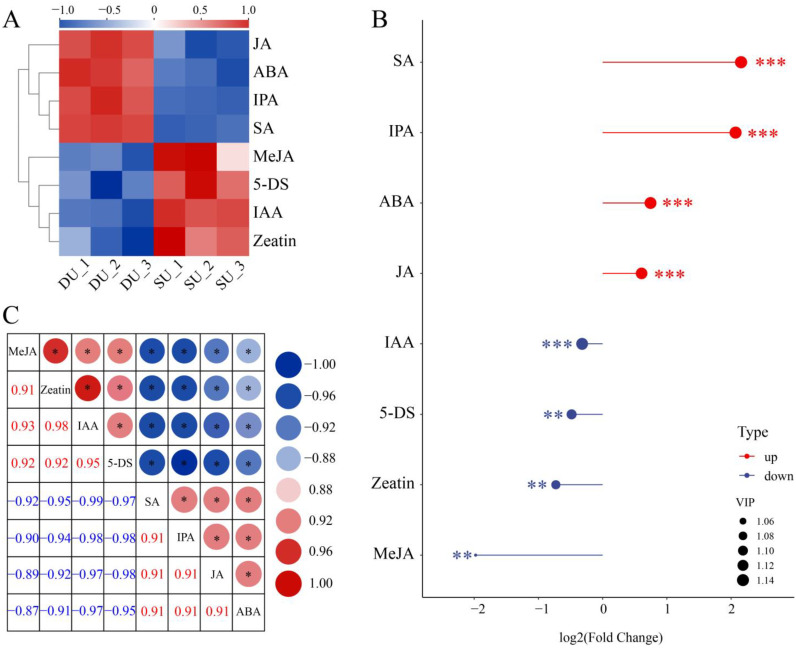
Analysis of differentially accumulated hormones in DU and SU groups. (**A**) HCA heatmap of DU vs. SU. The color blocks at different positions represent the relative expression of metabolites; red indicates high expression of the metabolite, and blue indicates low expression. (**B**) Matchstick analysis of DU vs. SU. The abscissa shows the log-transformed fold change, and the dot color represents the VIP value. **, 0.001 < *p* < 0.01; ***, *p* < 0.001. (**C**) The correlation analysis of DU vs. SU. The color blocks at different positions represent the magnitude of correlation coefficient between the metabolites, the red color indicates a positive correlation, and the blue color indicates a negative correlation. *, *p* < 0.05. SA, salicylic acid; IPA, isopentenyl adenosine; ABA, abscisic acid; JA, jasmonic acid; IAA, indole-3-acetic acid, 5-DS, 5-dexoxystrigol; MeJA, methyl jasmonate; SU, the top of the petiole in SB; DU, the top of the petiole in DB.

**Figure 5 ijms-25-06149-f005:**
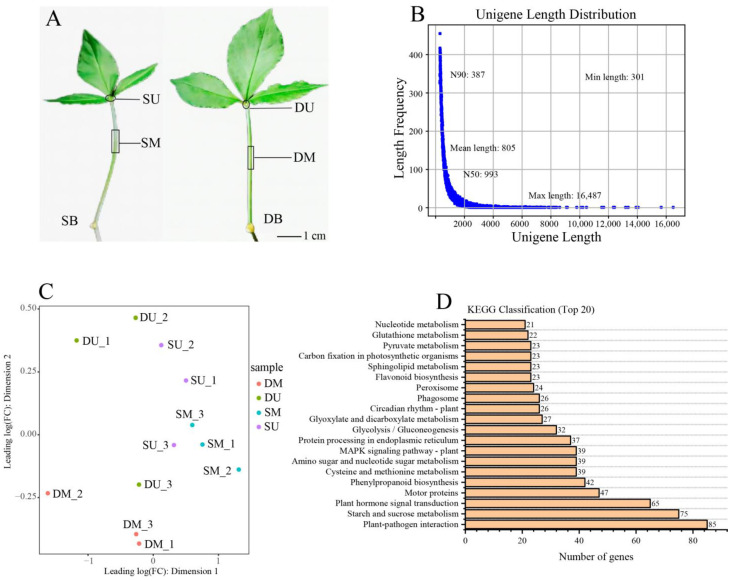
Transcriptome analysis of *P. ternata*. (**A**) Schematic representation of the transcriptomic sample positions: the top of the DB petiole, the middle of the DB petiole, the top of the SB petiole, and the middle of the SB petiole. (**B**) Unigene length distribution. (**C**) MDS Plot of DUs, SUs, DMs, SMs samples. (**D**) Classification of KEGG metabolic pathways. SU, the top of the petiole in SB; DU, the top of the petiole in DB; DM, the middle of the petiole in DB; SM, the middle of the petiole in SB; SB, single-bulbil type; DB, double-bulbil type.

**Figure 6 ijms-25-06149-f006:**
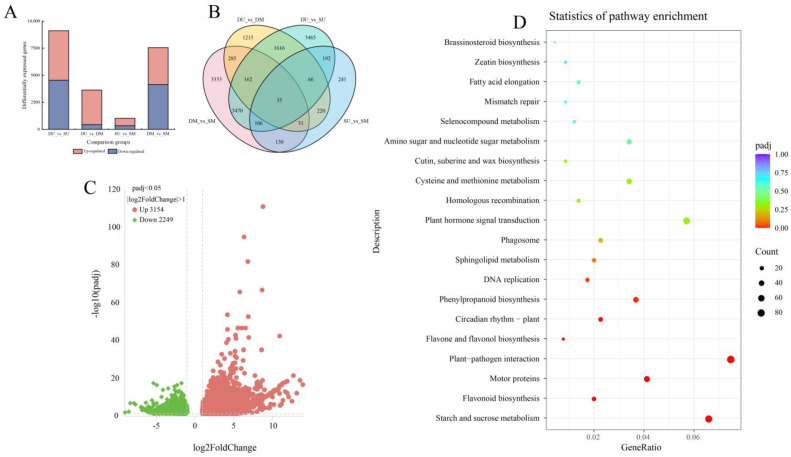
DEGs. (**A**) Number of up- and downregulated genes. (**B**) Venn diagram of differentially expressed genes in different comparison groups. (**C**) A volcano plot was generated to visualize the DEGs identified from DU vs. SU comparison. (**D**) KEGG pathways enriched in DEGs were identified from DU vs. SU comparison. SU, the top of the petiole in SB; DU, the top of the petiole in DB; DM, the middle of petiole in DB; SM, the middle of petiole in SB.

**Figure 7 ijms-25-06149-f007:**
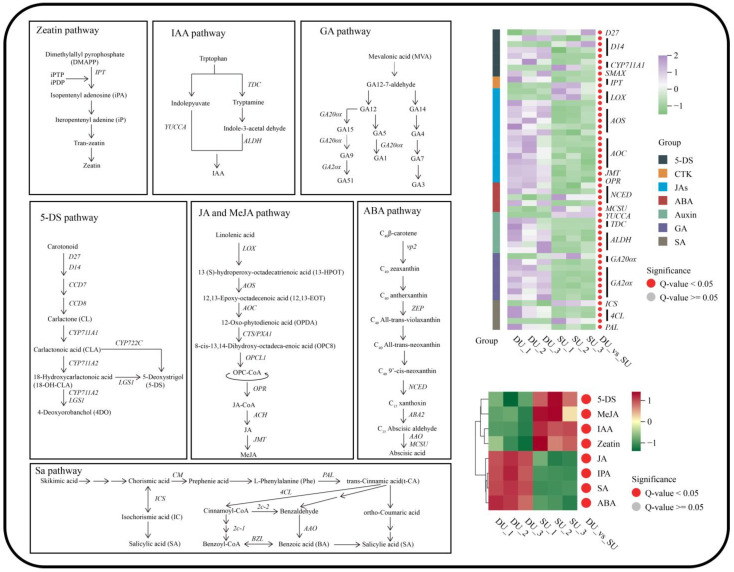
Analysis of 5-DS, CTK, JA, ABA, IAA, GA, and SA biosynthesis pathways in DUs and SUs. The pinkish-purple and green colors in the upper heatmap reflect the upregulated and downregulated gene expressions, respectively. Similarly, the red and green colors in the lower heatmap represent increased and decreased metabolite accumulation, respectively. SA, salicylic acid; IPA, isopentenyl adenosine; ABA, abscisic acid; JA, jasmonic acid; IAA, indole-3-acetic acid, 5-DS, 5-dexoxystrigol; MeJA, methyl jasmonate; SU, the top of the petiole in SB; DU, the top of the petiole in DB.

**Figure 8 ijms-25-06149-f008:**
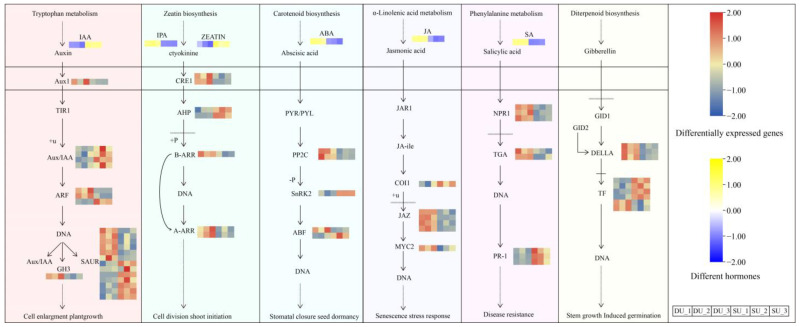
Analysis of hormone signaling pathway components in DU and SU. The heatmap illustrates the expression levels of genes and metabolites involved in the IAA, CTK, ABA, JA, SA, and GA signaling pathways. The red and blue colors represent upregulated and downregulated gene expressions, respectively, while the yellow and blue colors reflect increased and decreased metabolite accumulations, respectively. The three columns on the left represent DU-1, DU-2, and DU-3, and the three columns on the right represent SU-1, SU-2, and SU-3. SU, the top of the petiole in SB; DU, the top of the petiole in DB.

**Figure 9 ijms-25-06149-f009:**
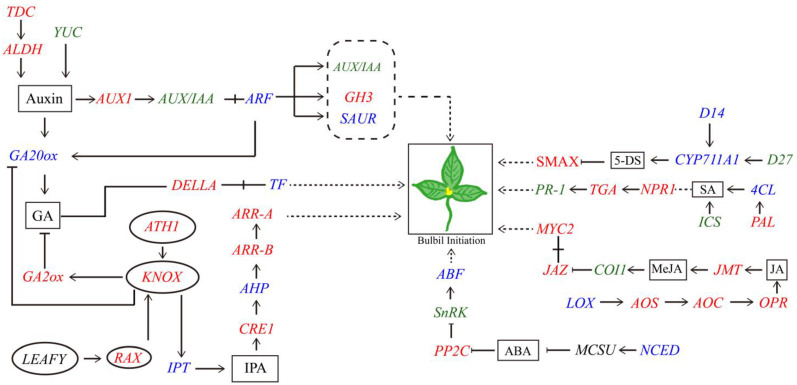
Proposed regulatory network of bulbil initiation in *P. ternata*. Hormones are designated with black boxes, while genes regulating AM are denoted by oval shapes. Genes involved in hormone synthesis and signal transduction pathways exhibit upregulation in red, downregulation in green, and both upregulation and downregulation in blue. The dashed arrow with a pointed tip signifies the presumptive positive control relationship, the pointed arrow represents positive control, and the flat arrow line with a blunt tip indicates an inhibitory effect.

**Figure 10 ijms-25-06149-f010:**
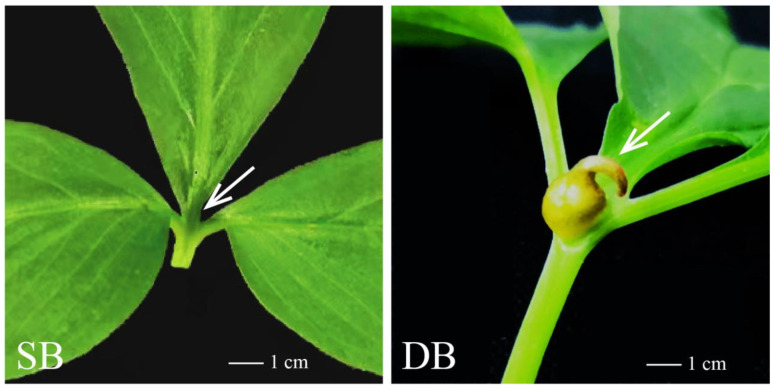
Overview of SB and DB in *P. ternata*. The rows indicate the top of the petiole. SB, single-bulbil type; DB, double-bulbil type. White arrowheads indicate different phenotypes at the top of the petiole with SB and DB. There is a bulbil at the top of the DB petiole, but none at the top of the SB petiole.

## Data Availability

The nucleotide sequence reported in this paper was submitted to the NCBI Sequence Reads Archive (SRA), accession number SRP484975.

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
