# Peer review of "Endogenous Hormone Levels and Transcriptomic Analysis Reveal the Mechanisms of Bulbil Initiation in Pinellia ternata"

_ijms, 2024, doi:10.3390/ijms25116149_

Round 1
Reviewer 1 Report
Comments and Suggestions for Authors
Dear Authors,
Thank you very much for your study. I reviewed your manuscript, and ı suggested some corrections and additions to important topics. After the revision, I think that it will be better compared to the first.
The suggestions:
1- Abstract Part: General information on Pinellia ternate should be given in the first paragraph of the Abstract section.
2- Line 63-66: Introduction part: The aim of the study should be given at the end of the introduction part combined with the line number 110-115.
3- Line 292-298: Results part: The subtitle “2.7. Verification of gene expressions” was explained so shortly. It should be rewritten again as clearly.
4- Discussion part: Why should the Discussion part be discussed under subheadings like in the results section? Discussing it under subheadings will produce clearer results.
4- Materials and Methods part: How were the applications made in the Materials and methods section determined? Please give relevant citations where necessary in all of the this part.
5- After third year, how was the natural withering determined, and how were healthy SB and DB corms collected? the authors should explain as clearly.
6- Conclusion part: The future impact of the study result should be clearly explained in the last paragraph.
General notes:
1- The manuscript should be revised and necessary corrections such as spelling errors and spaces between words should be made.
2- The references should be updated and give nearest time references studied as manuscript topic.
Sincerely yours,
Comments on the Quality of English LanguageThe language of the manuscript should be reviewed.
Reviewer 2 Report
Comments and Suggestions for Authors
Dear authors,
The authors present the morphological, hormonal, and transcriptional relationship as an important factor in bulbil initiation in Pinellia ternata. Various morphology, hormone, and transcriptome data were generated. Linking the data obtained in these three segments was attempted. In general, the manuscript and data analysis are poorly designed, data are poorly presented, and lacks rigor (both in morphological, hormonal, and transcriptional data), besides being confusing, with no fluidity, and difficult to understand. Data from related literature is practically not used and adequately discussed in the discussion section.
1. The manuscript contains several typing and English errors.
2. Scientific names should be in italic.
3. This abbreviation "zeatin(ZEATIN)" doesn't look good.
4. Introduction section, contains several sentences unreferenced and need to correct. Several extremely long and poorly prepared sentences. In certain places it is confusing and lacking in objectivity. Figure 1 should be used in the material and methods or results, and is not usual in the introduction section. Figure 1 needs to be a photo under a magnifying glass or microscope, to better highlight what you want to highlight.
5. Figures 2 and 3 are very low informative, generalist, non-explanatory, and non-critical. Figure 3 legend was not informed about "Figure 3a, d, g, i, m, p, s, v" or “Figure 3b, c, n, q” mentioned in the text. In general, bulbil morphology data are poor, a very long section with superficial information, non-critical, basic observations, and poorly supported observations. Figure 2 refers to SB or DB? Why not evaluate both? Figure 3 without visible dimension ruler.
6. In figure 4, the quantitative data about metabolites are omitted and statistical data was not provided. Phytohormone analysis, based on "expression" “down-regulated” did not sound good, since it can lead to genetic expression. Phytohormone data is poorly presented, poorly written, and very confusing. In Figure 5C, who are DU and SU?
7. Functional annotation and classification in the results section are poorly presented and written. Figure 6C, what explains in KEGG enrichment data, plant genes enriched pathways for "Sensory system, Nervous system, Excretory system, circulatory system, endocrine system, among others? What was the objective of comparing the SU versus SM and DU versus DM by RNAseq, in the context of the manuscript? The DEG analysis data are also poorly presented, poorly written, and very confusing. Some scant information is spoken but not shown. In particular, in DU vs SU, thousands of genes were differentially expressed, and the discussion about these is poor and has not addressed all possibilities. The association between hormone metabolites and transcriptomes mentioned only a few points. Several genes are mentioned as having a major effect, but no expression figures or tables are shown. Provide a table with gene expression profile for this sentence "The expression levels of KNOX, WUS , CLV, ATH1, SRL, MOC1, CUC, RAX, REV, and SPL were significantly greater in DUs than in SUs, indicating their upregulation during the top bulbil differentiation. Additionally, LEAFY and DA1 showed higher expression levels in SUs, suggesting their decreased expression during this developmental stage.", including other major genes involved in bulbil initiation of P. ternata. As well as for “Furthermore WUS, CLV, CUC, GA20ox, A-RRA, B-ARR, RAX, CRE, IPT, GH3, ARF, RAX, DAI were highly expressed and GA2ox, AUX/IAA were low expressed during this stage.” Where has RNAseq data related to SM and DM been used? KNOX gene expression data are not shown. Who are "the expressions of AOS and AOC ??
8. Figure 7B is merely illustrative, it is impossible to know how many genes are up- or down-regulated by looking at it. What is the difference and importance of Figure 6C compared to 7C? What is the difference between unigenes and differentially expressed genes, and why present both?
9. The discussion section is there is no rigor, poorly presented and written, poorly assembled, confusing, without objectivity, and lack of discussion of data with the literature. The following paragraph is duplicated in the discussion section: “Through morphological and histological observations, we determined that the optimal sampling period for studying bulbil initiation was prior to complete leaf unfurling. The initiation of bulbil started in the early plant growth stage, which does not require much nutrient accumulation from maternal plant. We, through three years of cultivation studies, conclude that the phenotype (DB/SB) can be stably inherited, and allexperiment materials obtained by tissue propagation from a DB or SB maternal plant showed consistent phenotypes of DB or SB, which is consistent with the previous report of stable bulbil phenotype of P. ternata [8]. Therefore, the bulbil initiation may be related to the ge-netic/germplasm material.”. Citations are deformatted and destandardized. The following similar and important published articles is fully unused and omitted: https://doi.org/10.3390/genes14091727, https://doi.org/10.3389/fpls.2021.809769, https://doi.org/10.3389/fpls.2023.1189499, as well https://doi.org/10.3390/horticulturae10020171 and https://doi.org/10.1186/s12870-022-03801-8
10. Figure legend are poorly elaborated and non-informative. Abbreviations are not explained.
11. The format of the text in the figure is not standardized.
12. What difference between "qRT‒PCR validation", "RT‒qPCR validation", "RT‒PCR validation", “qPCR”, and "real-time PCR validation"?
13. First mention of the term and its subsequent abbreviations are not uniform throughout the manuscript.
14. References for software used in bioinformatics are completely omitted.
15. The reference list is not in a unified format, it contains typing errors, scientific format errors, among others.
16. The following article is very poorly used in this manuscript: “Liu D., L. R. C. H. Members of PEBP Gene Family Involved in the Bulbil Development in Pinellia Cordata Revealed By Analysis of Transcriptome and Profiles, Journal of Guizhou University(Natural Sciences 2018,35(2),48-71.”
17. Validation of RNAseq data by RT-PCR is not presented and adequately addressed.
18. The following term was not found in SRA: SUB14162576.
19. Material and Methods: exaggeratedly long and with several typing and writing pattern errors. Are SB and DB different cultivars? Morphological of the top bulbil in DB of P. ternata. And for SB, in a comparative way? What is "IDU sample"?
Comments on the Quality of English Language1. The manuscript contains several typing and English errors.
Reviewer 3 Report
Comments and Suggestions for Authors
The manuscript presents a characterization of bulbil development using histological, transcriptomic and targeted metabolomic approaches in attempt to unravel the relationship between hormone, genes and bulbil development. The work is well presented and sufficiently supported, and may contribute with relevant biological information to the subject. Overall, I highlighted some minor concerns and suggested additional analyses to reinforce authors interpretations.
MATERIAL AND METHODS
In page 14, line 448 authors claim that only DU and SU were used for metabolomic analyses, however in lines 456-457 authors state that DU, SU, DM, and SM samples were ground using liquid nitrogen, and three biological replicates of each sample were included in the experiment. This point sounded confuse and should be clarified.
IDU sample (page 15, line 485) was not declared above. Is IDU sample the same as DU?
Authors did not mention the program used for raw read counts generation after transcriptome assembly and the type of data (whether raw count reads or CPM, FPKM normalized data) used for DEG by DESeq2.
Authors mention that primers sequences can be accessed at Table S3, but I found it in Table S11. This point must be adjusted.
RESULTS
Authors mention that the reference transcriptome was annotated with Swiss-Prot, NR, Pfam, KEGG, KOG, and GO databases, but they did not include nor describe it in the methodology. Also, I missed to see the annotation results using SwissProt and NR databases.
RT-qPCR validation shown in Fig S4 only exhibit RT-qPCR data. However, a convincing validated must present a comparison between RT-qPCR and RNA-seq expression results. Thus, I recommend to convert relative expression to log2FC and show both data together as Log2FC of similar comparisons. Also, I strongly suggest to perform person correlation analysis to demonstrate whether the expression pattern is conserved and RNA-seq experiment is reliable.
In order to highlight the association among gene expression levels of DEGs related to hormone and TFs; and among DEGs and hormone content I would like to see co-expression network interactomes created based on correlation analyses. No doubt, it would strongly reinforce authors interpretations and conclusions.
DISCUSSION
The third paragraph of discussion is a copy of the second.
Figure 10 should include an explanation of the regulatory network proposed so that it become clear and self-understandable. I think gene co-expression interactomes are more informative and convincing and could replace this figure.
Round 2
Reviewer 1 Report
Comments and Suggestions for Authors
Dear Aouthors,
Thank you very much for your revised manuscript. The manuscript was good now compared to old version.
Sincerely yours,
Author Response
Dear reviewer 1,
Thank you for your guidance on my manuscript. It is my pleasure to adhere to your requirements.
Kindly regards,
Lan Mou
Reviewer 2 Report
Comments and Suggestions for Authors
Dear authors of ijms-2933367,
The current version contains significant improvements. However, some specific points require important improvements.
1- The use of figures in the introduction is not wrong, but it is not common and is not standard, so it is not recommended, then remove.
2- The Materials and Methods section is exaggeratedly long and not very dense, so it is necessary to improve it.
3- Data shown in the body of the manuscript without statistics does not contribute for the reader's understanding, although supplementary material is used to provide support. In these cases, only one file (table or figure unified) must be generated showing both the data and statistical support in it, without the need to use several files to obtain the information. Statistics must appear in the same figure, either by using asterisks or letters that represent statistical significance or not.
4 - The morphology data and photos are of low informative quality (generalist, non-explanatory, and non-critical), so this needs and must be improved by showing what is really wanted clearly.
5 - The reference: "Liu D., L. R. C. H. Members of PEBP Gene Family Involved in the Bulbil Development in Pinellia Cordata Revealed By Analysis of Transcriptome and Profiles, Journal of Guizhou University(Natural Sciences 2018,35(2),48-71.”, it is not to be removed, but rather it must be explored rigorously.
6 - This reference "https://doi.org/10.3389/fpls.2024.1343222", must be explored rigorously.
7- A deep meta-analysis with the following studies x your data should be done:: https://doi.org/10.3390/genes14091727,
https://doi.org/10.3389/fpls.2021.809769, https://doi.org/10.3389/fpls.2023.1189499,
as well https://doi.org/10.3390/horticulturae10020171 and
https://doi.org/10.1186/s12870-022-03801-8
8 - The RNA-seq data were evaluated in a guided way to look at some genes, while other important ones may be omitted. Therefore, your discussion needs to have the following subsections, dissecting each of them and pointing out potential genes as involved in the phenotypic event: At least 1) Sucrose-triggered signaling pathway; 2) T6P-triggered signaling pathway; 3) Broad hormonal changes (ABA, JA, SA, ET, CK, AUX, BR, others); 4) Cytokinin signaling pathway; 5) Auxin signaling pathway; 6) Strigolactones signaling pathway; 7) Major genes and transcription factors involved in the phenotype; 8) Others...
9 - The discussion must compare its data and results with all related studies in the literature, until the content and information are exhausted, in order to make this section scientifically critical and of high level. The current section is still poor.
10 - PageMan and MapMan enrichment/categories analyses were not performed with the differentially expressed genes, but they must be done.
11 - In the gene expression data generated by RNA-seq, were FDR values used as a cutoff?
12 - Based on the "Sup.Table S12 A comparison between RT-qPCR and RNA-seq expression", a comparison using Pearson correlation coefficient should be performed for validation of data, gene per gene. You can use real-time RT-PCR values (2^-deltaCt) and TPM or FPKM values generated by RNA-seq for each library/sample.
13 - In this table "Sup.Table S13 The correlation of endogenous hormone and DEGs", should be provide the FDR values for support.
14 - The list of differentially expressed genes/transcripts identified in RNA-seq was not provided as supplementary files, it should also include p-value and FDR support. Please, provide this file.
15 - The results of "Figure S5. The Co-expression network of endogenous hormone and DEGs" are not properly discussed in the manuscript.
16 - The captions for supplementary figures and tables are either absent or poorly designed. Provide them with all the information necessary for understanding.
17 - In the introduction and discussion sections, there are numerous sentences or phrases of data from the literature that do not mention their respective references. This is not correct and all sentences must be properly referenced, meeting scientific standards.
18 - Not all supplementary material is properly cited or used in the manuscript. Please review this.
19 - The list of references was not standardized and does not have a DOI number.
20 - Heat maps need to be more informative containing statistical support in the same figure for each cell/character, including when comparing DU x SU.
21 - The representation of "Figure 10. Proposed regulatory network of bulbil initiation in P. ternata." is outdated because an analysis targeting only a few genes of interest was used. After analyzing the different avenues suggested for discussion, this data needs to be rigorously reviewed.
22 - The text still presents typing errors and unusual writing, which need to be corrected at this stage. Please, I'll do this.
23 - A MDS analysis of the RNA-seq datasets/libraries should be provided, with percentages represent variance captured by each principal component 1 and 2 in each analysis.
24 - A table with the gene name, gene ID, gene function, expression profile, and statistical support (p-value and FDR) for each/all gene identified as important for the phenotype must be provided in the manuscript.
25 - The chromosomal locations of the major genes associated with the phenotype (table suggested) should be provided to exploit eventual QTL.
26 - The Melting curve plot of gene amplification by real-time RT-PCR, for each gene, should be provided as supplementary data.
27 - Supplementary information about primer sequences should also provide information about melting temperature and amplicon size.
28 - A table with a summary of RNA libraries, sequencing raw data, filtered reads, and mapped reads for the samples was not provided. Please, provide it as a supplementary table.
29 - Figure 6C, the data was enriched for "sensory system, nervous system, excretory system, digestive system, others". Note that this process does not occur in plants. Please, explain about this.
Comments on the Quality of English LanguageNo comments.
Round 3
Reviewer 2 Report
Comments and Suggestions for Authors
Dear authors, thank you very much for the opportunity to review your manuscript.
7- A deep meta-analysis with the following studies x your data should be done: https://doi.org/10.3390/genes14091727, https://doi.org/10.3389/fpls.2021.809769, https://doi.org/10.3389/fpls.2023.1189499, as well https://doi.org/10.3390/horticulturae10020171, and https://doi.org/10.1186/s12870-022-03801-8
Response: It is not enough to mention them, they need to be analyzed, the data compared, and the implications discussed. It is not restricted to just these, but the literature needs to be reviewed.
-------------------------------------------------------
8 - The RNA-seq data were evaluated in a guided way to look at some genes, while other important ones may be omitted. Therefore, your discussion needs to have the following subsections, dissecting each of them and pointing out potential genes as involved in the phenotypic event: At least 1) Sucrose-triggered signaling pathway; 2) T6P-triggered signaling pathway; 3) Broad hormonal changes (ABA, JA, SA, ET, CK, AUX, BR, others); 4) Cytokinin signaling pathway; 5) Auxin signaling pathway; 6) Strigolactones signaling pathway; 7) Major genes and transcription factors involved in the phenotype; 8) Others...
Response: The study focuses on "Endogenous Hormone Levels and Transcriptomic Analysis Re-2 Veal the Mechanism of Bulb Initiation in Pinellia ternata", but the results and discussion do not dissect the hormonal pathways adequately and in depth. In fact, some of these topics are only marginally addressed, without depth, and a few genes are chosen among hundreds of DEGs to be considered as main in the "Figure 9. Proposed regulatory network of bulbil initiation in P. ternata.". Genes that make up the pathways or processes mentioned above need to be identified and the expression profile consulted in the DEGs list, then, the implications of these DEGs within each pathway need to be addressed, the relationship with the phenotype speculated, and discussed with support from the literature. MapMan results should also be used here. Finally, Figure 9 needs to be revised following these results.
-------------------------------------------------------------
10 - MapMan enrichment/categories analyses were not performed with the differentially expressed genes, but they must be done.
Response: Plant hormone pathways that are impacted by the differentially expressed genes should be shown by MapMan analysis. Provide an Excel table with follows collums: BinCode MapMan, BinName MapMan, Gene/Protein ID, Functional description, Log2 fold-change. Then, these data must be presented and then discussed.
------------------------------------------------------
12 - Based on the "Sup.Table S12 A comparison between RT-qPCR and RNA-seq expression", a comparison using Pearson correlation coefficient should be performed for validation of data, gene per gene. You can use real-time RT-PCR values (2^-deltaCt) and TPM or FPKM values generated by RNA-seq for each library/sample.
16 - The captions for supplementary figures and tables are either absent or poorly designed. Provide them with all the information necessary for understanding.
Response: How the Pearson correlation was calculated to validate RNA-seq was not described in detail in the material and methods and the table legend Table S7. In general, the legend of this and other tables is poorly designed. There is practically no information to understand what is being presented in the table. Provide all information in great detail, so that the caption is self-sufficient for understanding.
----------------------------------------------------------------
14 - The list of differentially expressed genes/transcripts identified in RNA-seq was not provided as supplementary files, it should also include p-value and FDR support. Please, provide this file.
24 - A table with the gene name, gene ID, gene function, expression profile, and statistical support (p-value and FDR) for each/all gene identified as important for the phenotype must be provided in the manuscript.
Response: Table S5 and Table S6 are polluted with data. Include an excel table with only the DEGs with only the following information: column 1: gene ID, column 2: logFC, 3: p-value, 4: FDR, 5: up or down, column 6: gene function/annotation. The list must contain all DEGs, not just some.
-----------------------------------------------------
23 - A MDS analysis of the RNA-seq datasets/libraries should be provided, with percentages represent variance captured by each principal component 1 and 2 in each analysis.
Response: MDS plot showing reproducibility among two biological replicates should be provided.
--------------------------------------------------------------------------------
Provide or justify why Strigolactones (SLs) and karrikins (KARs) were not evaluated in this study, as shown in figure 4. Since the focus is on hormones.
------------------------------------------
28 - A table with a summary of RNA libraries, sequencing raw data, filtered reads, and mapped reads for the samples was not provided. Please, provide it as a supplementary table.
Response: Include in Table S2, number and % of mapped reads to reference.
--------------------------------------------------
29 - Figure 6C, the data was enriched for "sensory system, nervous system, excretory system, digestive system, others". Note that this process does not occur in plants. Please, explain about this.
Response: The Arabidopsis gene ID (corresponding ortholog) must be added to the list of Unigenes, for each gene. Once we have the Arabidopsis gene ID, KEGG pathway enrichment will be conducted using the Arabidopsis dataset, but not with animal datasets. Figure 5C.
-------------------------------------------------
Comments on the Quality of English LanguageNo comments.
Author Response
Please see the attachmen!

Round 4
Reviewer 2 Report
Comments and Suggestions for Authors
Dear authors, thank you for the opportunity to read again your study. Thousands of genes were identified as DEGs and some pathways and processes were identified as contrasting, but only some limited were discussed. In particular, the subsections 3.5 to 3.8 are fully poorly elaborated. The discussion, mainly on these topics, is shallow. KEGG analysis showed that DEGs enriched several biological pathways. Each one of these enriched pathways should be addressed and deeply discussed in the discussion section.
Additional comments:
1. Again, the legends of supplemental figures and tables are poorly elaborated, lack minimal information, and overall contain only a short statement. Please, provide detailed information (assay, treatments, experimental design, statistics, replicates, explain the abbreviations, and several others). They can be evident to authors, but for readers, no.
2. In the Suppl. Table S14, melting temperature, provides the temperature used in the real-time RT-PCR assays or experiments. The temperature in silico predicted is not accepted.
3. In the Suppl. Table S7, comparison between RT-qPCR and RNA-seq expression: the first comparison should be using deltaCt values obtained by real-time RT-PCR and TPM or FPKM values obtained by RNA-seq for each gene in each library/sample. For example, find the deltaCt values of Cluster-8921.41248 gene in DU_1, in DU_2, in DU_3, in SU_1, in SU_2, and in SU_3 samples. Then, also find the TPKM or TPM values in DU_1, in DU_2, in DU_3, in SU_1, in SU_2, and in SU_3 libraries. Likewise for SM and DM. Lastly, should be carried out the Pearson correlation coefficient for each gene using these 6 deltaCt values and 6 values TPM values. You must use the deltaCt values, and no deltadeltaCt values. Here the aim is to validate the level of normalized transcripts for each gene in each sample/library. All this information must also be explained in the table legend. All legends of figures or tables must contain all the detailed information necessary for a reader who reads them for the first time to understand what is being presented. These results should be reported in the subsection 2.7.
4. The Suppl. Table S5 and S6, the list must contain all DEGs, not just some, or just related to hormones. All DEGs mentioned in the Fig. 6A.
5. The MDS plot graphs should be included within Figure 5. Instead of colored balls, use the name of each dataset within the figure, as it is difficult to differentiate one from the other. About the samples/libraries DU_1, DU_2, and DU_3: appear to be quite distinct, with DU_3 distant from the other 2 replicas. Please also analyze other replicates and explain.
6. Here, in the abstract, "Twelve transcription factors (TFs) involved in regulating bulbil initiation were identified. KNOX, WUS, CLV, ATH1, SRL, MOC1, CUC, RAX, REV, and SPL were highly expressed, while LEAFY and DA1 exhibited opposing expression patterns. In conclusion, utilizing the function of KNOX in regulating AM development", this sentence should be in accordance with the information presented in Figure 9.
7. Figure 1 has 9 images, but they were not explained in the legend. Please, insert Fig. 1a, 1b, 1c.... and explain each one.
8. In Figure 6, include in Venn diagram for DEGs in each sample or between samples.
9. Typing errors, for example, p < 0.05, is p-value? In addition, p-value should be in italics.
10. In the subsection 2.6, mixes results with discussion. The discussion should be in the Discussion section.
11. Figure 10, has a very low resolution and difficult to view the phenomenon.
12. In the conclusion section, this sentence should be in accordance with the information presented in Figure 9.
13. Make sure Figure 9 is properly updated with the information presented in subsection 2.6.
14. Legend of Figure 9, is poorly elaborated. The molecular mechanism needs to be described in detail and the abbreviations also must be described in the legend. What is the difference between dotted and non-dotted arrows? word with black font represents down-regulated genes?
15. Subsection 3.5, is poorly elaborated. Here, “we also identified 75 DEGs in the sucrose and starch metabolic pathways, among which 54 genes were upregulated”, This section should dissect these 75 DEGs and their implications for the phenomenon.
16. Subsection 3.6 is poorly elaborated. Please, deeply improve this topic about T6P. Some references, but non limited to these:
Ponnu, J., Wahl, V. & Schmid, M. (2011) Trehalose-6-phosphate: Connecting plant metabolism and development. Frontiers in Plant Science, 2, 70. https://doi.org/10.3389/fpls.2011.00070
Nunes, C., O’Hara, L.E., Primavesi, L.F., Delatte, T.L., Schluepmann, H., Somsen, G.W., Silva, A.B., Fevereiro, P.S., Wingler, A. & Paul, M.J. (2013) The trehalose 6-phosphate/SnRK1 signaling pathway primes growth recovery following relief of sink limitation. Plant Physiology, 162(3), 1720-1732. https://doi.org/10.1104/pp.113.220657
Paul, M.J., Gonzalez-Uriarte, A., Griffiths, C.A. & Hassani-Pak, K. (2018) The role of trehalose 6-phosphate in crop yield and resilience. Plant Physiology, 177(1), 12-23. https://doi.org/10.1104/pp.17.01634
Zhang, Y., Primavesi, L.F., Jhurreea, D., Andralojc, P.J., Mitchell, R.A.C., Powers, S.J., Schluepmann, H., Delatte, T., Wingler, A. & Paul, M.J. (2009) Inhibition of SNF1-related protein kinase1 activity and regulation of metabolic pathways by trehalose-6-phosphate. Plant Physiology, 149(4), 1860-1871. https://doi.org/10.1104/pp.108.133934
Wingler, A. & Henriques, R. (2022) Sugars and the speed of life - Metabolic signals that determine plant growth, development and death. Physiologia Plantarum, 174(2), e13656. https://doi.org/10.1111/ppl.13656
Morales-Herrera, S., Jourquin, J., Coppé, F., Lopez-Galvis, L., De Smet, T., Safi, A., Njo, M., Griffiths, C.A., Sidda, J.D., Mccullagh, J.S.O., Xue, X., Davis, B.G., Van der Eycken, J., Paul, M.J., Van Dijck, P. & Beeckman, T. (2023) Trehalose-6-phosphate signaling regulates lateral root formation in Arabidopsis thaliana. PNAS, 120(40), e2302996120. https://doi.org/10.1073/pnas.2302996120
Gazzarrini, S. & Tsai, A.Y.-L. (2014) Trehalose-6-phosphate and SnRK1 kinases in plant development and signaling: the emerging picture. Frontiers in Plant Science, 5, 119. https://doi.org/10.3389/fpls.2014.00119
Barbier, F.F., Cao, D., Fichtner, F., Weiste, C., Perez-Garcia, M.-D., Caradeuc, M., Le Gourrierec, J., Sakr, S. & Beveridge, C.A. (2021) HEXOKINASE1 signalling promotes shoot branching and interacts with cytokinin and strigolactone pathways. New Phytologist, 231(3), 1088-1104. https://doi.org/10.1111/nph.17427
Fichtner, F., Barbier, F.F., Annunziata, M.G., Feil, R., Olas, J.J., Mueller-Roeber, B., Stitt, M., Beveridge, C.A. & Lunn, J.E. (2021) Regulation of shoot branching in Arabidopsis by trehalose 6-phosphate. New Phytologist, 229(4), 2135-2151. https://doi.org/10.1111/nph.17006
Fichtner, F., Olas, J.J., Feil, R., Watanabe, M., Krause, U., Hoefgen, R., Stitt, M. & Lunn, J.E. (2020) Functional features of TREHALOSE-6-PHOSPHATE SYNTHASE1, an essential enzyme in Arabidopsis. The Plant Cell, 32(6), 1949-1972. https://doi.org/10.1105/tpc.19.00837
Lin, Q., Wang, J., Gong, J., Zhang, Z., Wang, S., Sun, J., Li, Q., Gu, X., Jiang, J. & Qi, S. (2023) The Arabidopsis thaliana trehalose-6-phosphate phosphatase gene AtTPPI improve chilling tolerance through accumulating soluble sugar and JA. Environmental and Experimental Botany, 205, 105117. https://doi.org/10.1016/j.envexpbot.2022.105117
Lin, Q., Yang, J., Wang, Q., Zhu, H., Chen, Z., Dao, Y. & Wang, K. (2019) Overexpression of the trehalose-6-phosphate phosphatase family gene AtTPPF improves the drought tolerance of Arabidopsis thaliana. BMC Plant Biology, 19, 381. https://doi.org/10.1186/s12870-019-1986-5
Avonce, N., Leyman, B., Mascorro-Gallardo, J.O., Van Dijck, P., Thevelein, J.M. & Iturriaga, G. (2004) The Arabidopsis trehalose-6-P synthase AtTPS1 gene is a regulator of glucose, abscisic acid, and stress signaling. Plant Physiology, 136(3), 3649-3659. https://doi.org/10.1104/pp.104.052084
17. In the subsection 3.7 are mentioned in poorly discussed only a restricted number of genes or TFs. Overall, the discussion section is poorly elaborated in the subsections about DEGs. It explores a limited number of information about and in a shallow way the DEGs data. It limits itself from the beginning to a small group of genes, and refuses to look at other mechanisms. The study showed thousands of DEGs, but here only a few are visited because they have a history in the literature.
18. Subsection 3.8, without sense and logic. DEGs and mechanisms related to this should be focused on this subsection. Then, these insights should be discussed with literature data.
Comments on the Quality of English LanguageNo comments.
Round 5
Reviewer 2 Report
Comments and Suggestions for Authors
Dear authors, thank you for the opportunity to read again your study. In my opinion, the current version needs minor corrections in English, but it is acceptable to publication after these careful corrections.
Comments on the Quality of English LanguageNo comments